# Temporo-parietal cortex involved in modeling one's own and others' attention

Arvid Guterstam[1,2]*, Branden J Bio[1], Andrew I Wilterson[1], Michael Graziano[1]

[1]Department of Psychology, Princeton University, Department of Psychology, Princeton, United States; [2]Department of Clinical Neuroscience, Karolinska Institutet, Stockholm, Sweden

**Abstract** In a traditional view, in social cognition, attention is equated with gaze and people track other people's attention by tracking their gaze. Here, we used fMRI to test whether the brain represents attention in a richer manner. People read stories describing an agent (either oneself or someone else) directing attention to an object in one of two ways: either internally directed (endogenous) or externally induced (exogenous). We used multivoxel pattern analysis to examine how brain areas within the theory-of-mind network encoded attention type and agent type. Brain activity patterns in the left temporo-parietal junction (TPJ) showed significant decoding of information about endogenous versus exogenous attention. The left TPJ, left superior temporal sulcus (STS), precuneus, and medial prefrontal cortex (MPFC) significantly decoded agent type (self versus other). These findings show that the brain constructs a rich model of one's own and others' attentional state, possibly aiding theory of mind.

## Introduction

Reconstructing someone else's attentional state is of central importance in theory of mind (*Baron-Cohen, 1997*; *Calder et al., 2002*; *Graziano, 2013*). By identifying the object of someone else's attention, and having some intuitive understanding of the complex dynamics and consequences of attention, one can reconstruct at least some of the other person's likely thoughts, intentions, and emotions, and make predictions about that person's behavior. Almost all work on how people reconstruct the attention of others has focused on gaze direction. For example, the human eye has a high contrast between pupil and sclera, possibly an adaptation for better gaze tracking (*Kobayashi and Kohshima, 1997*). The superior temporal sulcus in monkeys and humans may contain specialized neural circuitry for processing gaze direction (*Hoffman and Haxby, 2000*; *Marquardt et al., 2017*; *Perrett et al., 1985*; *Puce et al., 1998*; *Wicker et al., 1998*). Seeing a face gaze at an object automatically draws one's own attention to the object (*Friesen and Kingstone, 1998*; *Frischen et al., 2007*). These and other findings show the importance of reconstructing gaze direction in social cognition.

To be adaptive in aiding theory of mind, however, a model of attention should be far more than a vector indicating gaze direction. We previously suggested that the human brain constructs a rich, dynamic, and predictive model of other people's attention (*Graziano, 2019*; *Graziano, 2013*; *Graziano and Kastner, 2011*). The model should contain information about different types of attention, about the rapidity or sluggishness with which attention tends to move from item to item, about how external factors such as salience and clutter are likely to affect a person's attention, and about how attention profoundly affects thought, memory, and behavior. In the proposal, that deeper model is constrained by incoming information, including gaze direction. However, other cues can also constrain the model. People rely on the other person's body posture, on cues in the surrounding environment, on speech, and on social context. For example, blind people must be able to build models of other people's attention without seeing the other person's eyes. Likewise, during a phone

*For correspondence:
arvidg@princeton.edu

Competing interests: The authors declare that no competing interests exist.

conversation, we cannot see the other person and yet we intuitively understand whether that person is attending to what we have said or is distracted by her own words or by a salient event on her end of the line.

Several recent experiments provide evidence for an automatically constructed model of the attention of others that may go beyond merely registering gaze direction (*Guterstam et al., 2019*; *Guterstam and Graziano, 2020a*; *Kelly et al., 2014*; *Pesquita et al., 2016*; *Randall and Guterstam, 2020*; *Vernet et al., 2019*). For example, *Pesquita et al., 2016* found that when participants watch an actor in a video attending to an object, the participants implicitly distinguish between whether the actor's attention was drawn to the object exogenously (bottom-up or stimulus-driven attention), or whether the actor endogenously shifted attention to the object (top-down or internally driven attention). Exogenous and endogenous attention are the two principal ways in which selective attention moves between objects. They are emphasized in distinct cortical networks (the ventral and dorsal attention networks), and they influence the behavior of agents in profoundly different manners (*Corbetta et al., 2008*; *Corbetta and Shulman, 2002*; *Posner, 1980*; *Shulman et al., 2010*). The ability to distinguish between someone else's exogenous and endogenous attention is therefore one example of how people may construct a rich, dynamic, and useful model of other people's attention beyond merely encoding gaze direction or identifying the object of attention.

Inspired by the vignette-style tasks widely used in studies on theory of mind (*Fletcher et al., 1995*; *Gallagher et al., 2000*; *Happé, 1994*; *Saxe and Kanwisher, 2003*; *Vogeley et al., 2001*), in the present study, we used functional magnetic resonance imaging (fMRI) and multi-voxel pattern analysis (MVPA) to study brain activity in participants while they read brief stories about people's attention (*Figure 1*). Some of the stories implied that attention was being attracted exogenously ('Kevin walks into his closet and notices the bright red tie...') and some stories implied that attention was being directed endogenously ('Kevin walks into his closet and looks for the bright red tie...'). We also included analogous stories written in the first person, casting the subject of the experiment as the agent ('You walk into your closet and notice the bright red tie...'). The study therefore used a

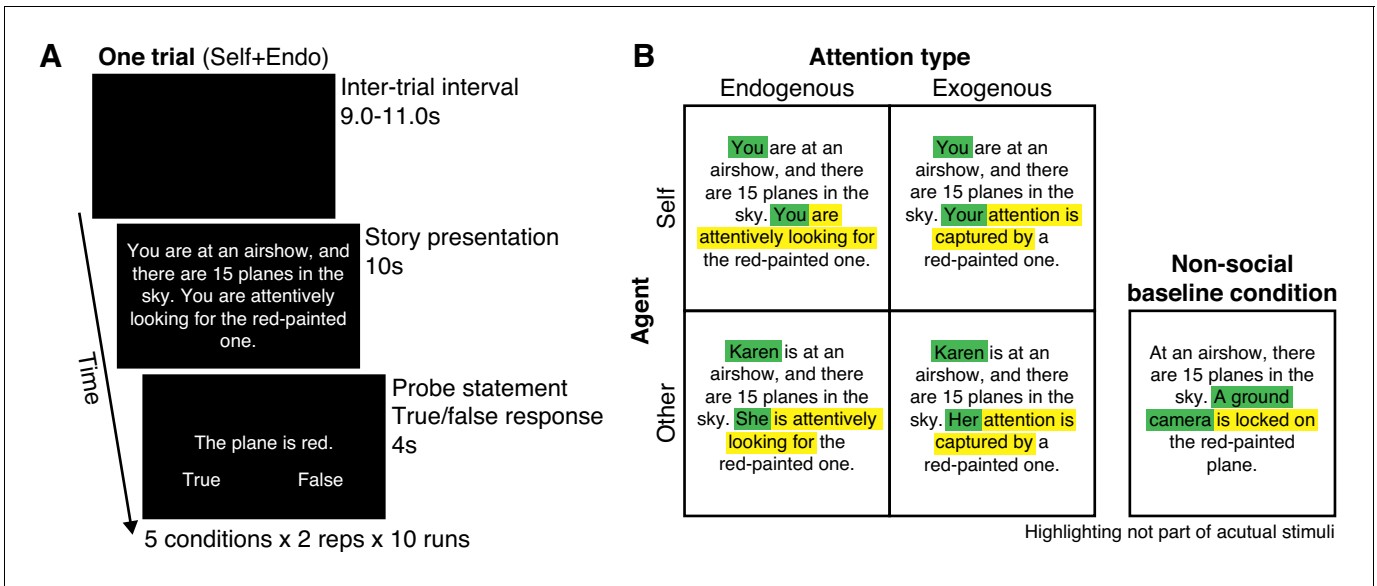

**Figure 1.** Methods. (**A**) Schematic timeline of the fMRI design. In each trial, subjects were presented with a short story for 10 s, describing a scene in which an agent attended to an object in the environment. A probe statement was then shown for 4 s, relating to either the story's spatial context or object property, to which the subjects responded either true or false by button press. (**B**) The agent in the story was either the subject him-/herself (self) or another person (other), and directed attention to the object endogenously (internally driven attention) or exogenously (stimulus-driven attention), yielding a 2 × 2 factorial design of attention type × agent. We created 80 unique stories in four different versions, one for each condition. We made minimal changes to the wordings to keep the story versions as semantically similar as possible. Green highlighting indicates wording specifying agent, yellow highlighting indicates wording specifying attention type (colors not part of actual visual stimuli). For each story, each subject saw only one of the four versions (balanced across subjects). We also included a nonsocial control condition (20 unique stories based on a subset of the 80 social stories) in which the agent was replaced by a non-human object.

2 × 2 design (exogenous versus endogenous attention X self agent versus other agent). Finally, we included a fifth, control condition, consisting of nonsocial stories in which the agent was replaced by an inanimate object, that, like attention, has a source and a target, such as a camera or a light source ('In a closet, a light shines on a red tie...').

We made four predictions. Our first, central prediction was inspired by the *Pesquita et al., 2016* study described above. We hypothesized that participants would encode the type of attention in the story (exogenous versus endogenous), and that this encoding would be evident in some subset of the areas classically involved in theory of mind. Previous experiments on theory of mind typically recruited a network of cortical areas including the temporoparietal junction (TPJ), the superior temporal sulcus (STS), the medial prefrontal cortex (MPFC), and the precuneus (*Gallagher et al., 2000*; *Saxe and Kanwisher, 2003*; *van Veluw and Chance, 2014*; *Vogeley et al., 2001*). This first prediction, that the social cognition network will encode the exogenous-versus-endogenous distinction, represents the main, novel contribution of this study. Previous studies have used MVPA to decode various aspects of other people's mental states from activity in social brain areas, such as their beliefs (*Koster-Hale et al., 2017*), intentions (*Koster-Hale et al., 2013*), and perceptual source (*Koster-Hale et al., 2014*). To the best of our knowledge, this investigation is the first to test whether activity in social brain areas can decode other people's attentional states.

Our second prediction was that participants would encode information about the agent in the story (self versus other), and that this encoding would again be evident in some subset of the areas classically involved in theory of mind. Self-versus-other encoding has been examined in previous studies, and found to be reflected in the theory-of-mind network (e.g. *Northoff et al., 2006*; *Ochsner et al., 2004*; *Passingham et al., 2010*; *Qin and Northoff, 2011*; *van Veluw and Chance, 2014*). This second prediction represents a test of whether our present paradigm, using subtle wording differences between similar sentences, can produce results consistent with previous findings.

Third, we predicted that participants would encode information associated with the interaction between the two factors. We predicted that at least some subset of the areas in the theory-of-mind network may encode the type of attention (exogenous versus endogenous) to a different extent in self-related stories as compared to other-related stories.

Fourth and finally, we tested for brain regions that encoded the distinction between social stories (with human agents) and nonsocial stories (with only non-agent objects). We predicted that this social-versus-nonsocial encoding would again be evident in the same network of brain regions noted above, that are known to be involved in theory of mind. This final analysis served as a control to check on the validity of the story stimuli and confirm that they engaged social cognition as expected.

## Results

### Behavioral results

Participants answered a simple true-or-false probe question after each story (e.g. 'Emma is on a bus'), to ensure alertness throughout the experiment. Although the MRI results depended on the time interval during the reading of the story and not during the reading or answering of the probe question, the behavioral response to the probe question may give some indication of whether the story conditions were well balanced. If one type of story was more difficult to process, or caused the subjects to think more deeply about the character in the story, that difference may be reflected in a different accuracy or latency in responding to the subsequent probe question. However, no significant differences were observed in accuracy ($F_{3,93}=0.15$, p=0.930, ANOVA) or latency ($F_{3,93}=1.66$, p=0.181, ANOVA) among the four social story conditions (*Table 1*).

When participants responded to the probe questions after non-social, control stories, versus when they responded to the probe questions after social stories, no significant difference in accuracy was found ($t_{31} = -1.83$, p=0.077), although as expected, a slightly longer latency was observed after non-social stories than after social stories ($t_{31} = 9.18$, p<0.001; see *Table 1*). The reason for the latency difference is almost certainly because the probe statements in the non-social condition were on average two words longer than those in the social conditions (eight words versus six words), because the character in the probe statements ("You ..." or "Emma ...") was replaced by an object that required more words to describe (e.g. "The surveillance camera ..."). One might therefore

**Table 1.** Behavioral results.
Mean accuracy and latency for the probe question that was presented after each story, for each of the five experimental conditions. The mean accuracy and latency across all social story conditions are also reported.

| | Story type | Self-endo | Self-exo | Other-endo | Other-exo | Non-social | Social (all) |
|---|---|---|---|---|---|---|---|
| Accuracy (%) | Mean | 94.2 | 93.4 | 93.1 | 93.4 | 90.0 | 93.6 |
| | SEM | 1.9 | 1.8 | 1.6 | 1.9 | 2.2 | 1.4 |
| Latency (ms) | Mean | 1613 | 1601 | 1667 | 1653 | 1957 | 1634 |
| | SEM | 54 | 56 | 46 | 56 | 63 | 49 |

expect that it took participants a little longer to read the non-social probe statements compared to the social ones. We suggest that this subtle difference in latency during the post-story question period is unlikely to have affected the comparison of MRI activity between social and non-social conditions, since the relevant MRI activity was evoked by the time period during the reading of the story, not during the reading and answering of the questions.

## Prediction 1

We hypothesized that participants would encode the attentional state of the agents in the stories in enough detail to distinguish between endogenous and exogenous attention, even though the difference between the story types was extremely subtle – only a few words that very slightly altered the semantic meaning of the sentences. We made the strong prediction that decoding would be found within the set of brain areas typically included in the theory-of-mind cortical network. *Figure 2* shows six ROIs within the theory-of-mind network, based on a meta-analysis of previous theory-of-mind studies (*van Veluw and Chance, 2014*). *Figure 3A* shows the results (see *Table 2* and *Figure 3—figure supplement 1* for more details). Decoding accuracy for endogenous versus exogenous stories was significantly above chance for the left TPJ, and the significance of the left TPJ decoding survived a multiple comparison correction for the six ROIs (mean decoding accuracy 52.9%, 95% CI 50.7–55.2, $p_{uncorrected}$ = 0.0046, $p_{FDR-corrected}$ = 0.0276). For the sake of a thorough evaluation, because different researchers have defined slightly different locations for the TPJ, we replicated the finding of a significant decoding in the left TPJ using three additional, previously reported theory-of-mind ROIs in the left TPJ (*Mar, 2011*; *Molenberghs et al., 2016*; *Schurz et al., 2014*), suggesting that the effect is robust (*Figure 3B* and *Figure 3—figure supplement 2*). The results therefore show that activity in the left TPJ allowed for significant decoding of the attentional state – exogenous versus endogenous – of agents in a story.

We then used a searchlight analysis over the whole brain to test whether any further areas may have significantly decoded the endogenous-versus-exogenous distinction. It should be noted that an

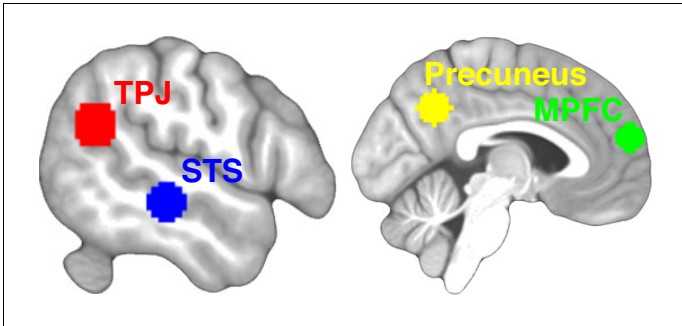

**Figure 2.** Regions of interest (ROIs). Six ROIs were defined based on peaks reported in an activation likelihood estimation meta-analysis of 16 fMRI studies involving theory-of-mind reasoning (*van Veluw and Chance, 2014*). The ROIs consisted of 10-mm-radius spheres centered on peaks in the bilateral temporoparietal junction (TPJ) and superior temporal sulcus (STS), and two midline structures: the precuneus and medial prefrontal cortex (MPFC). Here, the TPJ and STS ROIs on the left side are shown.

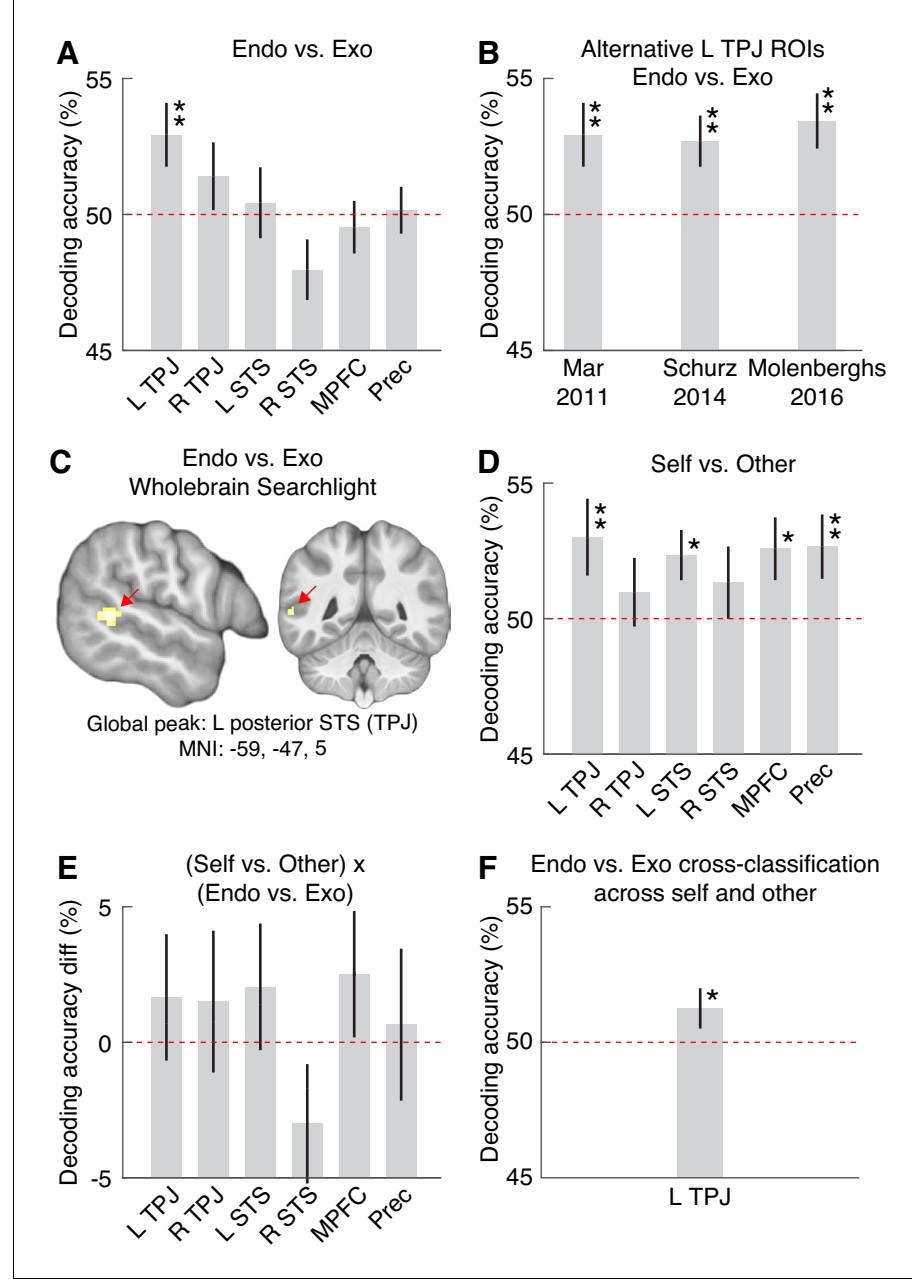

**Figure 3.** Decoding attention type, agent, and the interaction between them, in six brain areas. For definition of ROIs, see *Figure 2*. Each point shows mean decoding accuracy. Error bars show SEM. Red horizontal line indicates chance level decoding. Significance indicated by * (p<0.05) and ** (p<0.01), based on permutation testing (all significant p values also survived FDR correction for multiple comparisons across all six ROIs [all significant corrected ps <0.05]). (A) The ability of a classifier, trained on BOLD activity patterns within each ROI, to decode endogenous (endo) versus exogenous (exo) attention. (B) To test the robustness of the endo-versus-exo decoding in the left TPJ, we replicated the results in three ROIs derived from theory-of-mind neuroimaging meta-analyses (*Mar, 2011*; *Molenberghs et al., 2016*; *Schurz et al., 2014*) other than the one used for the main analysis (*van Veluw and Chance, 2014*) (see *Figure 3—figure supplement 2* for details). (C) The cluster shown had the highest decoding accuracy in the whole-brain, searchlight analysis, for the endo-versus-exo comparison. No clusters in this analysis survived brain-wide correction for multiple comparisons. We here report clusters surviving the conventional uncorrected voxelwise threshold p<0.001, for purely descriptive purposes. See *Figure 3—figure supplement 3* and *Supplementary file 1* for details. (D) Decoding accuracy for agent (self versus other). (E) Decoding accuracy for the interaction between type of attention and agent. (F) The ability of a classifier, trained to

*Figure 3 continued on next page*

*Figure 3 continued*

discriminate attention type in self stories, to decode attention type in other stories, and vice versa (i.e. two-way cross-classification), based on activity patterns in the left TPJ ROI.

The online version of this article includes the following figure supplement(s) for figure 3:

**Figure supplement 1.** Decoding attention type, agent, and the interaction between the two, within the six ROIs.
**Figure supplement 2.** Replicating attention type decoding results in additional TPJ ROIs.
**Figure supplement 3.** Decoding attention type at the whole-brain level.
**Figure supplement 4.** Decoding agent type at the whole-brain level.
**Figure supplement 5.** Decoding attention-by-agent interaction at the whole-brain level.
**Figure supplement 6.** Decoding attention type in the dorsal attention network.
**Figure supplement 7.** Decoding of attention type generalizes across self-stories and other-stories.
**Figure supplement 8.** Eye tracking results.
**Figure supplement 9.** Univariate fMRI results within ROIs.

exploratory searchlight analysis, compared to ROI analyses based on strong predictions, is much more statistically conservative because of the brain-wide correction for multiple comparisons. Its usefulness is that it may reveal any cluster of very strong decoding that was missed by the more sensitive analysis restricted to the ROIs. We found no brain-wide significant clusters of decoding for the endogenous versus exogenous distinction. However, four clusters survived the uncorrected p<0.001 threshold, and are reported in a purely descriptive manner in *Figure 3—figure supplement 3* and *Supplementary file 1*. The brain-wide searchlight peak was located in the left posterior STS (decoding accuracy 53.7%, t = 4.21, p<0.001 uncorrected; *Figure 3C*), at a distance of 20 mm from the center of the left TPJ ROI, and coincided with the posterior TPJ (TPJp) subregion as defined by *Bzdok et al., 2013* and *Mars et al., 2012*.

To control for potential univariate effects that could drive classifier performance, we explored the *endogenous > exogenous* and *exogenous > endogenous* univariate contrasts, which did not reveal any significant activity within the ROIs (*Figure 3—figure supplement 9*) or anywhere else in the brain, not even at the uncorrected threshold p<0.001 (*Supplementary file 5*). These findings are compatible with previous studies (e.g. *Hassabis et al., 2009*) that have demonstrated the superiority of pattern-sensitive multivariate analyses compared with conventional univariate approaches for detecting differences in activity between conditions with highly similar macroscopic characteristics.

In addition to these planned analyses, we explored the endogenous-versus-exogenous decoding within dorsal attention network regions. This analysis was motivated by an alternative hypothesis: people might simulate the act of attention orienting when reading the exogenous and endogenous stories, and thus activate the corresponding ventral (exogenous) attention network, to which the TPJ

**Table 2.** Decoding attention type, agent, and the interaction between the two, within the six ROIs.

For definition of ROIs, see *Figure 2*. Mean decoding accuracy (%), 95% confidence interval (based on bootstrap distribution), and p value (based on permutation testing) are shown for each of the six ROIs. Results shown for decoding endogenous (endo) versus exogenous (exo) attention type, self versus other agent type, and the interaction between the two variables. * indicates significant p values that survived correction for multiple comparisons across all six ROIs (FDR-corrected p<0.05).

| | | L TPJ | R TPJ | L STS | R STS | MPFC | Precuneus |
|---|---|---|---|---|---|---|---|
| Endo vs. Exo | Mean accuracy | 52.9% | 51.4% | 50.4% | 48.0% | 49.5% | 50.2% |
| | 95% CI | 50.7–55.2 | 49.1–53.9 | 47.8–52.8 | 45.9–50.1 | 47.6–51.4 | 48.5–51.8 |
| | P value | 0.0046* | 0.1148 | 0.3518 | 0.9547 | 0.6439 | 0.4428 |
| Self vs. Other | Mean accuracy | 53.0% | 51.0% | 52.3% | 51.3% | 52.6% | 52.7% |
| | 95% CI | 50.1–55.6 | 48.5–53.4 | 50.6–54.1 | 48.9–54.1 | 50.5–55.0 | 50.4–55.0 |
| | P value | 0.0053* | 0.1974 | 0.0204* | 0.1241 | 0.0105* | 0.0099* |
| (Self vs. Other) × (Endo vs. Exo) | Mean accuracy diff | 1.6% | 1.5% | 2.0% | −3.0% | 2.5% | 0.6% |
| | 95% CI | −2.7–6.3 | −3.6–6.5 | −2.7–6.3 | −7.4–1.1 | −2.3–6.7 | −5.5–5.5 |
| | P value | 0.2430 | 0.2639 | 0.1967 | 0.8944 | 0.1414 | 0.3900 |

belongs, and dorsal (endogenous) attention network in a 'mirror-neuron-like' fashion (see Discussion for details). However, this control analysis revealed no significant decoding in any of the dorsal attention network ROIs (*Figure 3—figure supplement 6*).

## Prediction 2

We hypothesized that participants would process the distinction between the two types of agent in the stories (self versus other). We made the strong prediction that decoding would be found within the same set of ROIs in the theory-of-mind cortical network. *Figure 3D* shows the results (see *Table 2* and *Figure 3—figure supplement 1* for more details). Decoding accuracy for self versus other stories was significantly above chance, and survived a multiple comparisons correction, for the left TPJ (mean decoding accuracy 53.0%, 95% CI 50.1 to –55.6, $p_{uncorrected} = 0.0053$, $p_{FDR-corrected} = 0.0210$), left STS (mean decoding accuracy 52.3%, 95% CI 50.6–54.1, $p_{uncorrected} = 0.0204$, $p_{FDR-corrected} = 0.0306$), MPFC (mean decoding accuracy 52.6%, 95% CI 50.5–55.0, $p_{uncorrected} = 0.0105$, $p_{FDR-corrected} = 0.0210$), and precuneus (mean decoding accuracy 52.7%, 95% CI 50.4–55.0, $p_{uncorrected} = 0.0099$, $p_{FDR-corrected} = 0.0210$). These results confirm that the present paradigm, using stories that are subtly different from each other, can obtain social cognitive results that are consistent with previous findings.

## Prediction 3

We hypothesized that areas in the theory-of-mind network would not only encode the distinction between endogenous and exogenous attention, but do so to a significantly different extent in self-related stories than in other-related stories. However, the results showed no significant interaction in any of the ROIs (*Figure 3E*, *Table 2*, and *Figure 3—figure supplement 1*). Thus, we found no support for prediction 3.

An alternative hypothesis is that attention type is encoded similarly in self and others. In a *post hoc* analysis, we focused on the left TPJ which was the only ROI that showed significant attention type decoding (*Prediction 1*), and tested for overlap in attention encoding in self and others using a two-way cross-classification analysis (see *Figure 3F* and *Figure 3—figure supplement 7*). In this analysis, one classifier was trained to discriminate endogenous versus exogenous self-stories and tested on other-stories, and another classifier was trained to discriminate endogenous versus exogenous other-stories and tested on self-stories, from which an average cross-classification decoding accuracy for the left TPJ was obtained. Endogenous-versus-exogenous decoding significantly generalized across self-stories and other-stories (mean decoding accuracy 51.9%, 95% CI 49.9–54.1, p=0.0393), suggesting at least some degree of overlap in the encoding of attention in others and in oneself.

## Prediction 4

Finally, we asked whether the activity in the theory-of-mind network would distinguish between social stories and nonsocial stories. This final analysis served as a control to check the validity of the story stimuli and confirm that they engaged social cognition as expected. We expected a signal of much greater magnitude in this analysis than in the analyses described above. The reason is that, as noted above, the types of social stories differed from each other by only a few words, and were nearly identical in semantic content; thus any brain signal reflecting those differences is expected to be subtle. The distinction between social and nonsocial stories, however, was much greater semantically, and therefore the evidence of decoding in the brain is expected to be of greater magnitude.

*Figure 4* shows the results (see *Table 3* and *Figure 4—figure supplement 2* for more details). The results are separated into six ROIs, and for each ROI, separated into four individual analyses, corresponding to each of the four main social conditions contrasted with the nonsocial control. Decoding accuracy was significantly greater than chance in almost all analyses across the six ROIs. The right STS showed the least consistent evidence of decoding. The TPJ bilaterally and the precuneus showed the most consistent evidence of decoding. These results show strong evidence of decoding of the social versus nonsocial stimuli in the known theory-of-mind, cortical network.

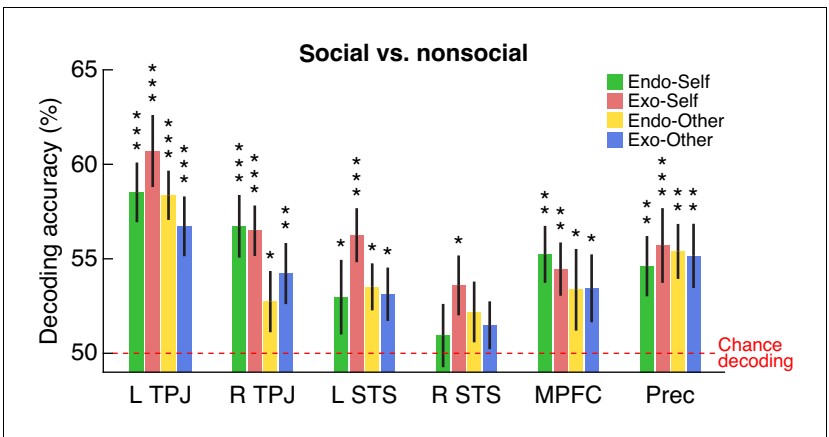

**Figure 4.** Decoding social versus nonsocial stories. The ability of a classifier, trained on BOLD activity patterns within each of the six ROIs, to decode each of the four social story conditions (endogenous-self, exogenous-self, endogenous-other, and exogenous-other) versus the nonsocial control. Each bar shows mean decoding accuracy, error bars show SEM, red horizontal line shows chance level decoding. Significance indicated by * (p<0.05), ** (p<0.01), and *** (p<0.001) based on permutation testing (all but one of the significant p values also survived FDR correction for multiple comparisons across all six ROIs; see *Table 3* for numerical details).
The online version of this article includes the following figure supplement(s) for figure 4:

**Figure supplement 1.** Decoding social versus nonsocial stories at the whole-brain level.
**Figure supplement 2.** Decoding social versus nonsocial stories within the six ROIs.

## Discussion

This study analyzed brain activity while people read stories about agents attending to objects in the environment. We examined whether specific brain areas could decode information about the type of attention referenced in the story (exogenous versus endogenous), and about the type of agent in the story (whether the agent was the subject reading the story or a different person). We hypothesized that if the brain constructs a model of attentional state that is used in social cognition, then areas of the brain known to be involved in social cognition should be able to distinguish between

**Table 3.** Decoding social versus nonsocial stories within the six ROIs.
For definition of ROIs, see *Figure 2*. Mean decoding accuracy (%), 95% confidence interval (based on bootstrap distribution), and p value (based on permutation testing) are shown for each of the six ROIs. Results shown for each of four social story conditions (endogenous-self, exogenous-self, endogenous-other, and exogenous-other) versus the nonsocial control. * indicates significant p values that survived correction for multiple comparisons across all six ROIs (FDR-corrected p<0.05).

|  |  | L TPJ | R TPJ | L STS | R STS | MPFC | Precuneus |
|---|---|---|---|---|---|---|---|
| Endo-Self vs. nonsocial | Mean accuracy | 58.5% | 56.7% | 53.0% | 50.9% | 55.2% | 54.6% |
|  | 95% CI | 55.5–61.6 | 53.5–59.8 | 49.3–56.8 | 47.5–54.0 | 52.3–58.0 | 51.5–57.7 |
|  | p value | 0.0001* | 0.0001* | 0.0338* | 0.2703 | 0.0026* | 0.0027* |
| Exo-Self vs. nonsocial | Mean accuracy | 60.7% | 56.5% | 56.3% | 53.6% | 54.5% | 55.7% |
|  | 95% CI | 57.0–64.4 | 53.7–58.8 | 53.5–59.0 | 50.5–56.6 | 51.7–57.1 | 51.9–59.4 |
|  | p value | 0.0001* | 0.0002* | 0.0001* | 0.0179* | 0.0044* | 0.0001* |
| Endo-Other vs. nonsocial | Mean accuracy | 58.4% | 52.7% | 53.5% | 52.2% | 53.4% | 55.4% |
|  | 95% CI | 55.9–60.9 | 49.6–55.9 | 51.1–55.9 | 48.8–55.0 | 49.5–57.7 | 52.6–58.2 |
|  | p value | 0.0001* | 0.0497 | 0.0147* | 0.0969 | 0.0219* | 0.0014* |
| Exo-Other vs. nonsocial | Mean accuracy | 56.7% | 54.2% | 53.1% | 51.5% | 53.4% | 55.2% |
|  | 95% CI | 53.8–59.8 | 51.1–57.3 | 50.4–55.9 | 49.1–54.0 | 49.8–56.7 | 52.0–58.5 |
|  | p value | 0.0002* | 0.0065* | 0.0319* | 0.1901 | 0.0197* | 0.0012* |

the two types of attention, exogenous and endogenous, represented in the stories. Our main analysis confirmed the hypothesis: the left TPJ showed significant decoding of information about endogenous versus exogenous attention. The finding is, arguably, remarkable, given that the semantic and wording difference between the two story types is extremely subtle.

These results support a new and growing body of evidence that the human brain constructs a model of attention to aid in theory of mind (*Guterstam et al., 2019*; *Guterstam and Graziano, 2020b*; *Kelly et al., 2014*; *Pesquita et al., 2016*; *Vernet et al., 2019*). The model includes information about attention that is deeper and more complex than just gaze direction or an identification of the attended object. At least one aspect of attention incorporated into the model appears to be the manner in which attention moves to an object: endogenously (internally directed) or exogenously (externally induced). The processing of the model appears to engage the theory-of-mind cortical network. The left TPJ showed the strongest decoding result.

Why should the TPJ in particular have shown involvement in decoding someone else's attention state, rather than areas in the STS that are known for responding to the gaze direction of others (*Marquardt et al., 2017*)? As noted in the Introduction, we suggest that modeling someone else's attention, and processing someone else's gaze direction, are not the same. Gaze is only one of many cues that can be integrated to constrain a model of someone else's attention. In *Kelly et al., 2014*, in a task in which participants judged the attention state of a cartoon, the TPJ was not active in association with gaze direction, and also not active in association with facial expression; but both left and right TPJ were active in association with the integration of the two cues, gaze and expression, in order to judge the cartoon's attentional state. The current finding of significant decoding in the left TPJ is consistent with that prior finding.

It is not clear why the left hemisphere showed stronger activity than the right in the present task. Social cognition tasks often activate the TPJ bilaterally, but typically engage the right TPJ more (*Saxe and Wexler, 2005*). One speculation is that some aspect of the present task, perhaps explicitly instructing people that the task was a test of reading comprehension, caused an emphasis on linguistic processing, biasing the activity toward the left hemisphere. Other explanations for the left-hemisphere bias may also be possible.

One alternative interpretation of the present finding in the TPJ is that people simulated attention orienting when reading the stories, and activated the corresponding dorsal (endogenous) and ventral (exogenous) attention networks in a 'mirror-neuron-like' fashion. Under this hypothesis, the TPJ, as part of the ventral attention network, was activated when reading the exogenous stories, and the dorsal attention network (consisting of the frontal eye fields, intraparietal sulcus, and middle temporal complex) should be active in association with reading the endogenous stories (*Corbetta et al., 2008*; *Corbetta and Shulman, 2002*; *Fox et al., 2006*). However, a control analysis showed that none of these dorsal attention network regions significantly decoded information concerning the attention type in the stories (*Figure 3—figure supplement 6*), which is incompatible with the attention-orienting simulation account of the TPJ result. Our findings are thus more consistent with the notion that the TPJ is involved in constructing a rich, perhaps implicit model of attention that may assist in social cognition (*Graziano and Kastner, 2011*; *Guterstam et al., 2020*; *Guterstam and Graziano, 2020a*; *Kelly et al., 2014*). We are not suggesting, however, that the possible involvement of the TPJ in modeling one's own and others' attention, and the involvement of the TPJ in the control of attention, is a coincidence. We suggested in previous work (*Graziano and Webb, 2015*) that a rich model of attention may be of benefit in the control of attention.

A second, alternative interpretation of the present results is that the exogenous sentences might make the reader focus more on the object in the story (thus engaging less mentalizing), whereas the endogenous sentences might make the reader focus more on the character's mental act of attending (thus engaging more mentalizing). In that interpretation, the TPJ shows a significant decoding result because it becomes more active in the endogenous story type, due to more mentalizing cognition. Although this possible explanation of the TPJ decoding results is difficult to exclude, we believe it is unlikely. First, if endogenous stories simply engaged more mentalizing than exogenous stories, and thus caused more activity in the TPJ, our univariate analysis should have found more activity in the TPJ to endogenous stories. It did not (see *Supplementary file 5*). We found no evidence that a simple difference in the amount of mentalizing between exogenous and endogenous stories affected the overall amount of activity in the TPJ or anywhere else in the brain. Although the pattern of activity in the left TPJ clearly contains information about the exogenous-endogenous distinction, that

information is not in the form of a simple increase in activity during the endogenous stories. Second, it is not clear that subjects should mentalize more in the endogenous story type than in the exogenous story type. When reading that 'Alice looks for the red ball,' subjects might wonder why, and think about Alice's mental state. When reading that 'Alice notices the red ball,' subjects might again wonder why the ball was of enough significance to be capturing her notice, or what mental reaction she experiences when noticing that apparently unanticipated object. Both story types invite mentalizing, although one focuses on an endogenous process of attention and the other on exogenous attention. Third, the probe task was designed to limit readers from speculating too much about the deeper meaning of the stories by asking only about literal details and never about the internal states of the characters. Thus, the subjects had an incentive to focus on the same physical aspects of the story in both the exogenous and endogenous conditions. Fourth, and finally, prior studies suggest that if an experimental and a control story both include human agents with potential thoughts, then subjects automatically mentalize in both story types, and the contrast between the story types will tend not to show a differential degree of activity in theory-of-mind cortical areas, especially in the TPJ (*Saxe and Kanwisher, 2003*). It is likely that, in our case, subjects cannot help mentalizing about the characters in both story types. We acknowledge that any story stimuli are always so complex that a variety of unintended, subtle differences may affect the results, but for the reasons listed here, we argue that a difference in the overall amount of mentalizing during endogenous versus exogenous stories is unlikely to explain the present results in the TPJ. The results point to the left TPJ processing different information in the exogenous and endogenous story types, but not a difference in overall amount of activity.

In addition to the endogenous-versus-exogenous comparisons, we also analyzed brain areas involved in self-versus-other encoding. We found evidence of self-versus-other encoding in the left TPJ, left STS, MPFC, and precuneus. The MPFC and precuneus have been previously implicated in self processing (*Northoff et al., 2006*; *Ochsner et al., 2004*; *Passingham et al., 2010*; *Qin and Northoff, 2011*; *van Veluw and Chance, 2014*), and the TPJ is consistently activated in fMRI studies involving self-recognition (*van Veluw and Chance, 2014*) and first-person perspective taking (*Ionta et al., 2011*). These results lend confidence to the present paradigm, showing that even the very subtle differences between our story stimuli were able to reveal cortical results consistent with previous studies.

Contrary to our prediction 3, we found no evidence for an interaction between attention type and agent type decoding in the theory-of-mind ROIs. (As noted in the *Supplementary Information*, during an exploratory searchlight analysis, we also found no evidence of an interaction effect in any other brain area.) Although it is possible that our paradigm was simply not sensitive enough to detect subtle interaction effects, these results suggest that the brain encodes information about attention type in a similar manner in the self and in others. In light of previous results showing that the attribution of sensory awareness to others and to oneself have a shared representation in the TPJ (*Kelly et al., 2014*), we directly tested this notion in a *post hoc* cross-classification analysis focusing on the left TPJ, which was the only region showing significant decoding of attention type. We found that endogenous-versus-exogenous decoding significantly generalized across self-stories and other-stories (*Figure 3—figure supplement 7*), suggesting that there is an overlap in brain mechanisms that participate in the encoding of attention in others and in encoding of attention in the self.

Finally, significant decoding of the social-versus-nonsocial distinction was obtained across most of the theory-of-mind ROIs. This finding confirmed the validity of the paradigm, and was expected based on previous experiments of the theory-of-mind network (*Gallagher et al., 2000*; *Saxe and Kanwisher, 2003*; *van Veluw and Chance, 2014*; *Vogeley et al., 2001*).

The use of a story-reading paradigm allowed us to systematically manipulate the kind of attention represented in the stimulus while keeping other experimental factors close to identical. The endogenous and exogenous story versions differed only with respect to a few key words specifying the type of attention, while the rest of the stories were semantically the same. The absence of any univariate effect within the ROIs or anywhere else in the brain, even at a liberal uncorrected threshold, confirm that the stimuli were well matched (*Supplementary file 5*). To avoid cognitive bias or expectation effects, the probe task performed by the subjects concerned details about the spatial context or the objects in the stories, effectively distracting subjects from the description of attention. We also speculate that this design of the probe task minimized the theoretical risk of the reader focusing on slightly different aspects of the story in the endogenous and exogenous conditions (e.g. focusing

more on the mental act of attending in endogenous stories, and more on the attention-grabbing object in the exogenous stories), because the task 'forced' the reader to focus on processing the spatial context and the object descriptions equally in both conditions. A post-scan questionnaire confirmed that none of the subjects came close to figuring out the purpose of the experiment (which they had been told was a 'Reading Comprehension Experiment'). The finding of brain areas that significantly decoded the type of attention, despite the distinction between endogenous and exogenous attention being subtle and task-irrelevant, suggests that the human brain automatically, and possibly also implicitly (*Pesquita et al., 2016*), constructs a model of an agents' attention that specifies at least some dynamic aspects of how that attention is moving around the scene.

## Materials and methods

### Subjects
Thirty-two healthy human volunteers (12 females, 30 right-handed, aged 18–52, normal or corrected to normal vision) participated in the study. Subjects were recruited either from a paid subject pool, receiving 40 USD for participation, or from among Princeton undergraduate students, who received course credits as compensation. In the subject recruitment material, the experiment was described as a 'Reading Comprehension Study.' All subjects provided informed consent and all procedures were approved by the Princeton Institutional Review Board.

### Experimental setup
Before scanning, subjects were instructed and then shown three sample trials (which were not part of the stories presented in the subsequent experiment) on a laptop computer screen. All subjects gave the correct response to all three trials on the first try, indicating they had understood the instructions adequately. During scanning, the subjects laid comfortably in a supine position on the MRI bed. Through an angled mirror mounted on top of the head coil, they viewed a translucent screen approximately 80 cm from the eyes, on which visual stimuli were projected with a Hyperion MRI Digital Projection System (Psychology Software Tools, Sharpsburg, PA, USA) with a resolution of 1920 × 1080 pixels. A PC running MATLAB (MathWorks, Natick, MA, USA) and the Psychophysics Toolbox (*Brainard, 1997*) was used to present visual stimuli. A right hand 5-button response unit (Psychology Software Tools Celeritas, Sharpsburg, PA, USA) was strapped to the subjects' right wrist. Subjects used the right index finger button to indicate a true response, and the right middle finger to indicate a false response during the probe phase of each trial.

### Experimental conditions and stimuli
Five experimental conditions were included. Subjects were presented with short stories (2–3 sentences, average word count = 24) describing a scene in which an agent, which was either the subject him-/herself (self) or another person (other), directed attention to something in the external world endogenously (e.g., 'X is attentively looking for Y') or exogenously (e.g., 'X's attention is captured by Y'). These four conditions made up a 2 × 2 factorial design: attention type (endogenous versus exogenous) X agent (self versus other). In addition, we included a control condition featuring stories in which the agent was substituted by a non-human object. In each trial, after a 9–11 s inter-trial interval, the story was presented for 10 s in easily readable, white text on a black background, at the center of the screen, after which a probe statement was shown for 4 s, to which the subjects responded either true or false by button press. See *Figure 1* for details, and *Supplementary file 6* for all stories.

Each subject ran 100 trials and thus saw 100 stories: 80 social stories and 20 non-social control stories. The 80 social stories were constructed as follows. We began with 80 unique short stories. For each story, four versions were constructed, one for each of the factorial conditions (*Figure 1B*). To keep the story versions as semantically similar as possible, we made minimal changes to the wordings. To distinguish the self and other versions, we substituted the word 'you' with a name (e.g. 'Karen') and the word 'your' with 'his' or 'her'. The names in the stories were selected from a list of the 100 most popular given names for male and female babies born during the years 1919–2018 in the United States, which is published by the Social Security Administration (https://www.ssa.gov/oact/babynames/decades/century.html). Half of the names were masculine, half feminine. To

distinguish the endogenous and exogenous story versions, we used different wording for the part of the story where the agent (X) is related to the object (Y). In the endogenous versions, we used formulations such as: 'X is trying to find Y,' 'X is trying to spot Y,' or 'X is looking attentively for Y.' In the exogenous versions, we used formulations such as: 'X's eyes are drawn to Y,' 'X's gaze is captured by Y,' or 'X's attention is captured by Y.' We matched the average number of words across all four conditions (24 words). The number of stories that included the words 'attention' or 'attentively' was balanced between the endogenous and exogenous categories (43 stories in each). Among the 80 stories, for each subject, 20 were randomly selected to be used in the endogenous-self version; 20 in the endogenous-other version; 20 in the exogenous-self version; 20 in the exogenous-other version. Thus, for the example story shown in *Figure 1B*, each subject saw only one of the four versions. In this manner, each subject saw 80 social stories, 20 of each type, balanced for as many properties as possible other than the two factors that were manipulated.

Finally, we constructed 20 additional stories for the non-social control condition (*Figure 1B*). To keep the control stories as semantically similar as possible to the social stories, we based them on a subset of the 80 original stories. Crucially, the agent in the original story was substituted with a non-human object, such as a camera or a spotlight, that has a source and a target just as attention does. For instance, the original story, "You are in a bike shop, and numerous bikes hang on one of the walls. You are attentively looking for that red Italian sports bike", was adapted to the non-social condition by substituting the agent with a spotlight: "In a bike shop, on one of the walls, hangs numerous bikes. A bright spotlight is shining on a red Italian sports bike". The average number of words of the non-social stories (24 words, standard deviation = 3) was matched with the attention stories.

The purpose of the probe statement at the end of each trial was to ensure that subjects carefully read the stories. Each statement described one detail of the preceding story that could be either true or false. We restricted the probe statements to the spatial context of the story (place probe: e.g. 'Emma is on a bus') or the object being described (object probe: e.g. 'The Van Gogh painting has sunflowers') in order to avoid alerting subjects to the focus of the experiment on theory of mind and attention. Half of the probe statements were place probes and half object probes. Within both the place and the object probes, half were true and half were false. The probe was on screen for 4 s, during which subjects were required to indicate whether the statement was true or false by button press.

The experiment consisted of 10 runs of approximately 4 min each. In each run, the five conditions were repeated two times, yielding a total of 10 trials per run. The trial order was randomized, with the limitation that two consecutive trials could not belong to the same condition. Each run included 18 s of baseline before the onset of the first trial and 12 s of baseline after the offset of the last trial.

## Post-scan questionnaire

At the end of the scanning session, subjects were asked what they thought the purpose of the experiment was and what they thought it was testing.

## fMRI data acquisition

Functional imaging data were collected using a Siemens Prisma 3T scanner equipped with a 64-channel head coil. Gradient-echo T2*-weighted echo-planar images (EPI) with blood-oxygen dependent (BOLD) contrast were used as an index of brain activity (*Logothetis et al., 2001*). Functional image volumes were composed of 54 near-axial slices with a thickness of 2.5 mm (with no interslice gap), which ensured that the entire brain excluding cerebellum was within the field-of-view in all subjects ($54 \times 78$ matrix, 2.5 mm x 2.5 mm in-plane resolution, TE = 30 ms, flip angle = 80°). Simultaneous multi-slice (SMS) imaging was used (SMS factor = 2). One complete volume was collected every 2 s (TR = 2000 ms). A total of 1300 functional volumes were collected for each participant, divided into 10 runs (130 volumes per run). The first three volumes of each run were discarded to account for non-steady-state magnetization. A high-resolution structural image was acquired for each participant at the end of the experiment (3D MPRAGE sequence, voxel size = 1 mm isotropic, FOV = 256 mm, 176 slices, TR = 2300 ms, TE = 2.96 ms, TI = 1000 ms, flip angle = 9°, iPAT GRAPPA = 2). At the end of each scanning session, matching spin echo EPI pairs (anterior-to-posterior and posterior-to-anterior) were acquired for blip-up/blip-down field map correction.

## FMRI preprocessing

Results included in this manuscript come from preprocessing performed using FMRIPREP version 1.2.3 (*Esteban et al., 2019*) (RRID:SCR_016216), a Nipype (*Gorgolewski et al., 2011*) (RRID:SCR_002502) based tool. Each T1w (T1-weighted) volume was corrected for INU (intensity non-uniformity) using N4BiasFieldCorrection v2.1.0 (*Tustison et al., 2010*) and skull-stripped using antsBrainExtraction.sh v2.1.0 (using the OASIS template). Spatial normalization to the ICBM 152 Nonlinear Asymmetrical template version 2009c (*Fonov et al., 2009*) (RRID:SCR_008796) was performed through nonlinear registration with the antsRegistration tool of ANTs v2.1.0 (*Avants et al., 2008*) (RRID:SCR_004757), using brain-extracted versions of both T1w volume and template. Brain tissue segmentation of cerebrospinal fluid (CSF), white-matter (WM) and gray-matter (GM) was performed on the brain-extracted T1w using fast (*Zhang et al., 2001*) (FSL v5.0.9, RRID:SCR_002823).

Functional data was slice time corrected using 3dTshift from AFNI v16.2.07 (*Cox, 1996*) (RRID:SCR_005927) and motion corrected using mcflirt (FSL v5.0.9) (*Jenkinson et al., 2002*). This was followed by co-registration to the corresponding T1w using boundary-based registration (*Greve and Fischl, 2009*) with six degrees of freedom, using flirt (FSL). Motion correcting transformations, BOLD-to-T1w transformation and T1w-to-template Montreal Neurological Institute (MNI) warp were concatenated and applied in a single step using antsApplyTransforms (ANTs v2.1.0) using Lanczos interpolation.

Many internal operations of FMRIPREP use Nilearn (*Abraham et al., 2014*) (RRID:SCR_001362) principally within the BOLD-processing workflow. For more details of the pipeline see https://fmriprep.readthedocs.io/en/latest/workflows.html.

## Testing prediction 1

The purpose of the first analysis was to determine whether the brain encoded information concerning the type of attention (endogenous or exogenous) present in the stories. For this analysis, we used MVPA, which tests whether patterns of brain activity can be used to decode the distinction between two conditions. It is a more sensitive analysis than the more common, simple subtraction methods. The reason for using this sensitive measure is that the difference between exogenous and endogenous trial types was extremely subtle. Both trial types engaged social cognition, and therefore might cancel each other out in a simple subtraction. The stimuli were nearly identical, differing only in a few words that indicated the type of attention used by the agent in the story. In addition, the type of attention featured in the story was irrelevant to the task performed by the subject. To accommodate the subtlety of the distinction between conditions, we designed the study to use MVPA. We hypothesized that with MVPA, brain activity would carry information about the endogenous versus exogenous distinction; and that decoding would be evident in regions of interest (ROIs) within the network of areas typically found to be involved in social cognition.

We defined our ROIs as spheres centered on the statistical peaks reported in an activation likelihood estimation (ALE) meta-analysis of 16 fMRI studies (including 291 subjects) involving theory-of-mind reasoning (*van Veluw and Chance, 2014*), in accordance with generally accepted guidelines in ROI analysis (*Poldrack, 2007*). The ROIs are shown in *Figure 2*. The peaks were located in six areas: the left TPJ (Montreal Neurological Institute [MNI]: −52,−56, 24), right TPJ (MNI: 55,−53, 24), left STS (MNI: −59,−26, −9), right STS (MNI: 59,−18, −17), MPFC (MNI: 1, 58, 19), and the precuneus (MNI: −3,−56, 37). The radius of the ROI spheres was 10 mm, corresponding to the approximate volume (4000 mm$^3$) of the largest clusters (TPJ and MPFC) reported in *van Veluw and Chance, 2014*. The same sphere radius was used for all ROIs.

The fMRI data from all participants were analyzed with the Statistical Parametric Mapping software (SPM12) (Wellcome Department of Cognitive Neurology, London, UK) (*Friston et al., 1994*). We first used a conventional general linear model (GLM) to estimate regression (beta) coefficients for each individual trial (i.e. 100 regressors), focusing on the 10 s story presentation phase of each trial. One regressor of no interest modeled the 4 s probe statement phase across conditions. Each regressor was modeled with a boxcar function and convolved with the standard SPM12 hemodynamic response function. In addition, ten run-specific regressors controlling for baseline differences between runs, and six motion regressors, were included. The trialwise beta coefficients for the endogenous and exogenous conditions (i.e. 80 beta maps) were then submitted to subsequent multivariate analyses (*Haxby et al., 2001*).

The MVPA was carried out using The Decoding Toolbox (TDT) version 3.999 (*Hebart et al., 2014*) for SPM. For each subject and ROI, we used linear support vector machines (SVMs, with the fixed regularization parameter of C = 1) to compute decoding accuracies. To ensure independent training and testing data sets, we used leave-one-run-out cross-validation approach. For each fold, the betas across all training runs were normalized relative the mean and standard deviation, and the same Z-transformation was applied to the betas in the left-out test run (*Misaki et al., 2010*). An SVM was then trained to discriminate activity patterns belonging to the endogenous or exogenous trials in nine runs, and then tested on the left-out run, repeated for all runs, resulting in a run-average decoding accuracy for each ROI and subject.

For statistical inference, the true group mean decoding accuracy was compared to a null distribution of group mean accuracies obtained from permutation testing. The same MVPA was repeated within each subject and ROI using permuted condition labels (10,000 iterations). A p value was computed as (1+the number of permuted group accuracy values > true value)/(1+the total number of permutations). To control for multiple comparisons across the six ROIs, we used the false discovery rate (FDR) correction (*Benjamini and Hochberg, 1995*). In addition, we also computed a bootstrap distribution around the true group mean accuracy by resampling individual-subject mean accuracies with replacement (10,000 iterations), from which a 95% confidence interval (CI) was derived (*Nakagawa and Cuthill, 2007*). A corrected p value < 0.05 in combination with a 95% CI that does not cross chance level were interpreted as a significant decoding effect at the group level (*Nakagawa and Cuthill, 2007*).

In addition, as further exploratory statistics beyond the targeted hypotheses of this study, we used a whole-brain searchlight analysis (*Kriegeskorte et al., 2006*) to test for possible areas of decoding outside the ROIs. This searchlight analysis is described in the *Supplementary Information* (*Supplementary file 1–4* and *7*, *Figure 3—figure supplements 3–5*, and *Figure 4—figure supplement 1*).

## Testing prediction 2

The purpose of the second analysis was to determine whether the brain encoded information concerning the type of agent (self versus other) present in the stories. The analysis methods were the same as for testing hypothesis 1, except that for regressors of interest we used the self-related and other-related trials, collapsed across the type of attention (exogenous or endogenous). Just as for hypothesis 1, we tested the six defined ROIs within the theory-of-mind network.

## Testing prediction 3

The purpose of the third analysis was to test for an interaction between the two variables (endogenous versus exogenous, and self versus other). We used MVPA to test whether the decoding for the type of attention was significantly different between the self-related and the other-related stories. The analysis methods were similar to those used for testing hypotheses 1 and 2, except in the following ways. We computed two MVPA decoding results, the first for distinguishing endogenous-self from exogenous-self stories, the second for distinguishing endogenous-other from exogenous-other stories. We then computed the difference between the two decoding results ([endogenous-self versus exogenous-self] – [endogenous-other versus exogenous-other]) to create a decoding difference score. Just as for hypotheses 1 and 2, we tested the six defined ROIs within the theory-of-mind network (*van Veluw and Chance, 2014*).

In a post hoc analysis, we tested for overlap in attention type encoding in self and others in the left TPJ ROI by using a two-way cross-classification analysis. One classifier was trained to discriminate endogenous versus exogenous self-stories and tested on other-stories (using a leave-one-run-out approach), and another classifier was trained to discriminate endogenous versus exogenous other-stories and tested on self-stories, from which an average cross-classification decoding accuracy was obtained. Overlap in attention type representations in self and other should be reflected in above-chance decoding.

## Testing prediction 4

The purpose of the fourth analysis was to confirm whether our story stimuli engaged social cognition and thereby recruited brain areas within the expected theory of mind network. The analysis was

meant as an added control to check the validity of the paradigm. The analysis methods were similar to those used for testing hypotheses 1–3, except in the following ways. We computed four MVPA decoding results: endogenous-self versus nonsocial, endogenous-other versus nonsocial, exogenous-self versus nonsocial, and exogenous-other versus nonsocial. (Because using MVPA to compare two conditions requires equal numbers of trials in both conditions, it was not possible to use a single analysis to compare all 80 social trials to the 20 nonsocial trials.) Each analysis represents a separate, alternative way to assess the social-versus-nonsocial decoding. Just as for hypotheses 1–3, we tested the six defined ROIs within the theory-of-mind network (*van Veluw and Chance, 2014*).

### Eye tracking analysis

Eye movements were recorded via an MRI-compatible infrared eye tracker (SR Research EyeLink 1000 Plus), mounted just below the projector screen, sampling at 1000 Hz. Before each scanning session, a calibration routine on five screen locations was used and repeated until the maximum error for any point was less than 1°. The obtained eye position data was cleaned of artifacts related to blink events and smoothed using a 20 ms moving average. We then built an SVM decoding model analogue to the cross-validation approach used for the fMRI data, but here based purely on eye tracking data, to test whether eye movement dynamics alone were sufficient to decode the conditions of interest (endogenous versus exogenous, and self versus other). In keeping with a previous study (*Schneider et al., 2013*), we organized the data in the following way. The part of the display within which the stimuli appeared was divided into an 8 × 4 grid of 32 equally sized squares. The grid covered the screen area within which the stories were presented (see red outline in *Figure 3—figure supplement 8*), and approximately corresponded to the locations of individual words (four lines, with eight words per line). For each trial, the proportion of time that the subject fixated within each square (32 features) and the saccades between those regions (32 × 32 = 1024 features) was calculated. These 1056 features, representing information about both where people were looking as well as saccade dynamics, were then averaged across repetitions for each of the four main conditions within each of the 10 runs, yielding one eye movement feature vector per condition per run (per subject). The feature vectors were submitted to an SVM classifier (C = 1). Using a leave-one-run-out approach, the SVM model was trained on endogenous versus exogenous story types, and then tested in the left-out run. At the group level, the decoding accuracies were tested against chance level using t-tests. A similar analysis was then performed on the contrast between self-related stories versus other-related stories. The results showed that endogenous-versus-exogenous and self-versus-other story types could not be decoded significantly better than chance using the pattern of eye movement. See *Supplementary Information* (*Supplementary file 7* and *Figure 3—figure supplement 8*) for the results of the eye-tracking analysis.

## Acknowledgements

This work was supported by the Princeton Neuroscience Institute Innovation Fund. Arvid Guterstam was supported by the Wenner-Gren Foundation, the Sweden-America Foundation, the Swedish Brain Foundation, and the Promobilia Foundation. The authors would like to thank Sam Nastase for valuable input regarding the multivoxel pattern analysis.

## Additional information

### Funding

| Funder | Author |
| --- | --- |
| Wenner-Gren Foundation | Arvid Guterstam |
| Swedish Brain Foundation | Arvid Guterstam |
| Sweden-America Foundation | Arvid Guterstam |
| Promobilia Foundation | Arvid Guterstam |
| Princeton Institute for International and Regional Studies | Arvid Guterstam<br>Branden J Bio<br>Andrew I Wilterson |

Michael Graziano

The funders had no role in study design, data collection and interpretation, or the decision to submit the work for publication.

## Author contributions
Arvid Guterstam, Conceptualization, Formal analysis, Funding acquisition, Investigation, Visualization, Methodology, Writing - original draft; Branden J Bio, Andrew I Wilterson, Investigation, Writing - review and editing; Michael Graziano, Conceptualization, Writing - review and editing

## Author ORCIDs
Arvid Guterstam (iD) https://orcid.org/0000-0002-3694-1318

## Ethics
Human subjects: All subjects provided informed consent and all procedures were approved by the Princeton Institutional Review Board (IRB# 10740).

## Decision letter and Author response
Decision letter https://doi.org/10.7554/eLife.63551.sa1
Author response https://doi.org/10.7554/eLife.63551.sa2

# Additional files

## Supplementary files
• Supplementary file 1. Decoding endogenous versus exogenous at the whole-brain level. All clusters (≥10 voxels) of decoding activity passing the voxelwise threshold of p<0.001 (none of the clusters survived p<0.05 correction for multiple comparisons using the whole brain as search space). *The left posterior STS cluster also coincided with the posterior TPJ (TPJp) subregion as defined by *Bzdok et al., 2013*; *Mars et al., 2012*.

• Supplementary file 2. Decoding agent type at the whole-brain level. All clusters (≥10 voxels) decoding agent type (self versus other) activity at the threshold of p<0.001 (uncorrected). Corrected p values represent cluster-level correction using the whole brain as search space.

• Supplementary file 3. Decoding attention-by-agent interaction at the whole-brain level. All clusters (≥10 voxels) in which endogenous-versus-exogenous decoding was better in self-related compared to other-related stories at p<0.001 (uncorrected). (none of the clusters survived correction for multiple comparisons using the whole brain as search space).

• Supplementary file 4. Decoding social versus nonsocial stories at the whole-brain level. Clusters (≥10 voxels) decoding social versus nonsocial stories significantly better than chance. The listed clusters represent the overlap of significant clusters (p<0.05, corrected using a cluster-defining uncorrected threshold of p<0.001 and the entire brain as search space) across four separate whole-brain searchlight analyses: endogenous-self versus nonsocial, exogenous-self versus nonsocial, endogenous-other versus nonsocial, and exogenous-other versus nonsocial.

• Supplementary file 5. Univariate fMRI results. All clusters (k ≥ 10) of voxels surviving a p<0.001 (uncorrected) threshold are shown for the univariate contrasts corresponding to the main multivariate analyses (*Figures 3–4*). The preprocessed functional images were smoothed using a 6 mm full FWHM Gaussian kernel and subjected to conventional GLMs. We report the p values for clusters that survived a cluster-level familywise error rate correction for multiple comparisons, either using the whole brain or one of the ROI volumes as search space.

• Supplementary file 6. Story stimuli. List of all of the stories and story versions (endogenous-self, exogenous-self, endogenous-other, and exogenous-other) presented to the participants during the experiment. Story #1–80 are the social stories, and #81–100 are the nonsocial control stories.

• Supplementary file 7. Searchlight analysis and eye-tracking decoding methods text.

• Transparent reporting form

### Data availability

The data that support the findings of this study are available at https://figshare.com/s/c3463d15bc78106a1b5c (https://doi.org/10.6084/m9.figshare.12273128).

The following dataset was generated:

| Author(s) | Year | Dataset title | Dataset URL | Database and Identifier |
|-----------|------|---------------|-------------|-------------------------|
| Guterstam A | 2021 | Temporo-Parietal Cortex Involved in Modeling One's Own and Others' Attention | https://doi.org/10.6084/m9.figshare.12273128.v1 | figshare, 10.6084/m9.figshare.12273128.v1 |

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
