## [Decision Letter]

**Acceptance summary:**

This paper reports the novel finding that fMRI activity patterns in the left TPJ distinguish between stories in which an agent endogenously versus exogenously attends to an object. The experimental design is straightforward and elegant, using tightly-controlled comparisons. The findings suggest that brain regions implicated in theory-of-mind represent a model of another person's attentional state.

**Decision letter after peer review:**

Thank you for submitting your article "Temporo-Parietal Cortex Involved in Modeling One's Own and Others' Attention" for consideration by *eLife*. Your article has been reviewed by three peer reviewers, one of whom is a member of our Board of Reviewing Editors, and the evaluation has been overseen by Michael Frank as the Senior Editor. The following individual involved in review of your submission has agreed to reveal their identity: Anthony Atkinson (Reviewer #3).

The reviewers have discussed the reviews with one another and the Reviewing Editor has drafted this decision to help you prepare a revised submission.

We would like to draw your attention to changes in our revision policy that we have made in response to COVID-19 (https://elifesciences.org/articles/57162). Specifically, when editors judge that a submitted work as a whole belongs in *eLife* but that some conclusions require a modest amount of additional new data, we are asking that the manuscript be revised to either limit claims to those supported by data in hand, or to explicitly state that the relevant conclusions require additional supporting data.

Summary:

This fMRI study tested whether theory-of-mind (ToM) regions differentially represent information about endogenous and exogenous attention of the self and of another person. Participants read short passages that described themselves or another person deliberately attending to or having their attention drawn to an object. The main finding was that activity patterns in left TPJ allowed for decoding endogenous vs. exogenous attention (collapsed across self and other). This finding indicates a role for the left TPJ in modelling attentional states.

Motivation and interpretation:

1) Considering that the study decodes mental states (here: attentional states) in ToM areas, as described in stories, please review work similarly decoding mental states in these regions (e.g., work by Koster-Hale, Saxe). How is your study similar or different from previous work decoding mental states? Do results follow from these previous studies? I also missed a review of (and integration with) the literature on endogenous vs exogenous attention itself (e.g., ventral vs dorsal attention network).

2) It was not clear from the Introduction why you hypothesized that exo vs endo attention should be decodable in TPJ. Is this where eye gaze direction – which appeared to have motivated the study – can be decoded from? The study cited for this hypothesis (Kelly et al., 2014; reanalysed in Igelstrom et al., 2016) showed that univariate activity in TPJ reflected the difficulty of social attribution, which does not obviously lead to the current hypothesis.

3) Does the main finding reflect a model of others' attention or differences in mental state attribution: sentences describing endogenous attention focus the reader more on the mental act of attending and might also induce further mentalizing (e.g., the reader may wonder: "why would he decide to look for that object?"). In exogenous sentences, the focus of the sentence is instead more strongly on the attention-grabbing object (e.g., "the bright red tie"). Can this alternative interpretation be excluded?

4) Searchlight analysis for exogenous v. endogenous attention: Is the cluster centred at -59, -47, 5, labelled left posterior STS (TPJ), really TPJ? It is, after all, squarely in the temporal lobe and some distance (22mm) from the centre of the left TPJ ROI, the latter being in left angular gyrus (areas PGa and PFm). Do you have some independent justification for the labelling of the location of this cluster as TPJ? For example, perhaps it lies within the anterior TPJ (TPJa) subregion identified by Mars et al. (Cerebral Cortex 2012)? The posterior STS area from the searchlight analysis seems to fall into the "gaze-following patch" in the posterior STS as reported by Marquardt et al., 2017. Since this cortical area is not generally considered to be a part of the theory of mind network, the seeming involvement of this area potentially changes the interpretation of the results.

Additional analyses:

5) Please report and compare accuracy and RTs for the response to the probe statement. If either differs between conditions, then that becomes a potential confound for the between condition contrasts in the MVPA analyses, considering that the BOLD response to the story and probe events could not be separated.

6) The TPJ is introduced in the context of theory of mind and social cognition, but it has also been implicated in attentional orienting. Could it be the case that participants simulate such orienting when reading the stories, leading to the above-chance decoding in TPJ? If this is the case, one may similarly expect above-chance decoding in areas that have been implicated in endogenous attention. This should be tested by including dorsal attention network ROIs. Above-chance decoding in such attention regions may inform the interpretation of the TPJ results.

7) Please provide univariate activity estimates, both in the ROIs and in whole-brain contrasts. This may help to interpret the multivariate results.

Further support for main finding:

8) The reported decoding accuracy values are really quite small and close to 50% (chance), especially for the endogenous vs. exogenous and self vs. other contrasts, even those values that are statistically significant. Please report appropriate effect sizes and the confidence intervals around those effect sizes (for all your reported t-tests, not just those that are statistically significant). It would also be informative if you were to include in your graphs the individual subject data (e.g., mean decoding accuracy per subject for each ROI) and/or plots of the effect sizes and their distributions. For more on effect sizes, their CIs and associated plots, I point you to the following sources and the references therein:

https://thenewstatistics.com/itns

https://thenewstatistics.com/itns/2019/05/20/reply-to-lakens-the-correctly-used-p-value-needs-an-effect-size-and-ci/

https://www.estimationstats.com/

9) The authors computed the attentional state decoding separately for "self" and "other". These decoding accuracies did not differ. Please also provide and test the accuracies of self and other separately; this would shed light on the reliability of the main result (which was collapsed across the two conditions) and might also indicate whether the decoding was more reliable (less variable) in self or other.

10) Replicate results in one or multiple additional ToM TPJ ROIs. Reviewers raised two suggestions: 1) Surface-based ROI: the TPJ is a highly anatomically variable cortical region that isn't well approximated by a single volume ROI (see Croxson et al., 2017). There are many surface-based ROIs now available that mitigate the effect of this variability, including ones from the authors' own lab. 2) Use a different meta-analysis: The used meta-analysis defines “theory of mind” solely in terms of false-belief tasks (van Veluw and Chance, 2014), which may not be appropriate for delineating the ROIs and their exact locations in your study. Different types of ToM task reliably activate different brain areas, as well as common ones (mPFC and bilateral TPJ: Molenberghs et al., Neuroscience and Biobehavioral Review 2016). Consider using the results of a different meta-analysis, e.g., one that identifies regions based on the conjunction of multiple types of theory-of-mind tasks (e.g., Mar, Annual Review of Psychology 2011; Molenberghs et al., 2016; Schurz et al., Neuroscience and Biobehavioral Reviews 2014).

11) In previous work (Kelly et al., 2014), the authors were interested in an overlap between attention in self and other. Here, this could be addressed in a cross-decoding analysis, training a classifier on exo vs endo in the "self" stories and testing this on the "other" stories. Above-chance classification in this analysis would strengthen the evidence for lTPJ involvement and would provide additional information that would help interpreting the TPJ findings.

[Editors' note: further revisions were suggested prior to acceptance, as described below.]

Thank you for submitting your article "Temporo-Parietal Cortex Involved in Modeling One's Own and Others' Attention" for consideration by *eLife*. Your article has been reviewed by two peer reviewers, one of whom is a member of our Board of Reviewing Editor, and the evaluation has been overseen by Michael Frank as the Senior Editor. The reviewers have opted to remain anonymous.

Summary:

Both reviewers noted that many of their concerns were addressed. However, they each had one remaining concern that would need to be addressed.

Essential Revisions:

1) Please report and analyze the univariate activity in the ROIs

2) More fully report (in main Results section) and discuss the searchlight results.

Reviewer #1:

The authors have addressed many of my concerns. In particular, the additional TPJ ROIs and the demonstration of self-other cross-decoding increased my confidence in the main finding of attention state decoding in lTPJ.

One comment has not been sufficiently addressed, however: it would be highly relevant and informative to see the average univariate activity for each of the conditions in the 6 original ROIs, plotted in a graph. The endo>exo contrast should be tested, correcting for the 6 ROIs, similar to how the decoding accuracies are tested (i.e., with the same p<0.05 cut-off). This ROI-based univariate analysis is also interesting for other comparisons, e.g. to test whether the social conditions gave higher activity than the non-social condition, as would be expected based on previous work. Thus, please provide a full univariate analysis on average ROI activity.

There is a lot of interesting information in the Supplementary figures (e.g., searchlight, additional ROIs), which will not be visible when viewing or printing pdfs. You could consider moving some of this to the main text, or add panels to existing figures where you include some of this information.

Reviewer #2:

In this revised manuscript, the authors have greatly clarified some key points, most important of which was how well matched the behavior was across all of the conditions. While there is still a potential confound in the social vs. non-social contrast given the difference in reaction time, (a) the confound effects are likely negligible in size, (b) the end result matches the previous literature, and (c) the contrast was only a control analysis and doesn't greatly affect the main point of the paper. However, the handling of the left posterior STS result in the searchlight analysis remains problematic, both in the manuscript and in the author's response.

pSTS response and discussion: The main point of the manuscript is to examine whether known theory of mind areas contain information that differentiates between the endogenous vs. exogenous story conditions. The revised manuscript makes quite clear that the answer is yes, activity in the L TPJ differentiates between the two conditions, albeit with a relatively low decoding accuracy. The fact that multiple versions of the L TPJ ROI produces the same result is particularly reassuring. The spotlight analysis, however, finds that there is potentially more discriminatory information to be found in the posterior STS/middle temporal gyrus, and this is where the manuscript runs into problems.

In the revised manuscript, this result is simply not mentioned, though it clearly has implications for the interpretations of the result as mentioned in the previous round of reviews. In addition, in the response to the reviewers' comments, the authors give a very convoluted and confusing argument based on MNI coordinates that somehow this focus is somehow (a) part of the TPJ and (b) may be a part of the attention-related TPJa as opposed to their TPJp ROI. In the process of saying that the surface projections are misleading, they state that the focus as falling on the STG on the surface projection, when it is really on the MTG. In addition, they base their arguments on MNI coordinates, but MNI coordinates are notoriously inaccurate in comparing results from different studies, so the author's attempts to justify their conclusions using these coordinates falls flat. The authors own volume images very clearly show the focus as being in the STS and not on either the angular gyrus or supramarginal gyrus that Mars, Corbetta, and others have generally shown the TPJp and TPJa to fall on. And even if this focus was in the TPJa, as the authors seem to hint at, they do not discuss the implications of this result.

The authors' defensiveness around this point is frankly puzzling. The searchlight results do not seem to invalidate their main point, only potentially augment it. The fact that the posterior STS (and what seem to be the right FEF and iPCS areas) exhibit higher discriminability between the endo and exo conditions may not be shocking given that the theory of mind areas are likely involved in many operations in this task, whereas the pSTS and right hemisphere areas likely may only be involved in just the imagined attention/sensory processing aspects of the task. It is possible that these areas are "reading out" the differing attentional conditions from the L TPJ. Whatever the explanation may be, the authors seem to be trying to bury this result to shoehorn the results into fitting their a priori model, which is a disservice and misleading to the readers, and needs to be rectified before the manuscript can be published. My recommendation is to at least mention the searchlight results in the main Results section, then add a paragraph to the Discussion discussing the implications of these results. Better yet would be to quantify the discrimination accuracy within each of the foci uncovered in the spotlight analyses.

---

## [Author Response]

Revisions for this paper:Motivation and interpretation:1) Considering that the study decodes mental states (here: attentional states) in ToM areas, as described in stories, please review work similarly decoding mental states in these regions (e.g., work by Koster-Hale, Saxe). How is your study similar or different from previous work decoding mental states? Do results follow from these previous studies? I also missed a review of (and integration with) the literature on endogenous vs exogenous attention itself (e.g., ventral vs dorsal attention network).

We thank the reviewers for these relevant suggestions. The work in our lab has focused on how people construct models of their own and others’ attention, an area of modeling mental states that is relatively less well studied. We now cite and briefly review previous relevant work using MVPA to decode various aspects of other people’s mental state, such as their beliefs (Koster-Hale et al., 2017), intentions (Koster-Hale et al., 2013), and perceptual source (Koster-Hale et al., 2014), from activity patterns in the theory of mind network. We also clarify what we believe distinguishes the current study from earlier ones. From the fifth paragraph of the Introduction:

“This first prediction, that the social cognition network will encode the exogenous-versus-endogenous distinction, represents the main, novel contribution of this study. Previous studies have used MVPA to decode various aspects of other people’s mental states from activity in social brain areas, such as their beliefs (Koster-Hale et al., 2017), intentions (Koster-Hale et al., 2013), and perceptual source (Koster-Hale et al., 2014). To the best of our knowledge, this investigation is the first to test whether activity in social brain areas can decode other people’s attentional states.”

We also added a brief review of endogenous versus exogenous attention in the revised Introduction (paragraph three):

“For example, Pesquita et al. (2016) found that when participants watch an actor in a video attending to an object, the participants implicitly distinguish between whether the actor’s attention was drawn to the object exogenously (bottom-up, or stimulus-driven attention), or whether the actor endogenously shifted attention to the object (top-down, or internally driven attention). Exogenous and endogenous attention are the two principal ways in which selective attention moves between objects. They may be relatively emphasized in distinct cortical networks (the ventral and dorsal attention networks), and they influence the behavior of agents in profoundly different manners (Corbetta et al., 2008; Corbetta and Shulman, 2002; Posner, 1980; Shulman et al., 2010). The ability to distinguish between someone else’s exogenous and endogenous attention is therefore one example of how people may construct a rich, dynamic, and useful model of other people’s attention beyond merely encoding gaze direction or identifying the object of attention.”

2) It was not clear from the Introduction why you hypothesized that exo vs endo attention should be decodable in TPJ. Is this where eye gaze direction – which appeared to have motivated the study – can be decoded from? The study cited for this hypothesis (Kelly et al., 2014; reanalysed in Igelstrom et al., 2016) showed that univariate activity in TPJ reflected the difficulty of social attribution, which does not obviously lead to the current hypothesis.

We thank the reviewers for allowing us to clarify the rationale behind our central *Prediction 1*. First, we would like to stress that the wording of *Prediction 1* emphasized that we expected above-chance decoding in “some subset” of theory-of-mind areas, not only in the TPJ:

“We hypothesized that participants would encode the type of attention in the story (exogenous versus endogenous), and that this encoding would be evident in some subset of the areas classically involved in theory of mind. […] We therefore predicted that the exogenous-versus-endogenous distinction would be significantly encoded in some subset of these areas.”

In accordance with this prediction, we tested exogenous-versus-endogenous decoding in each of the six ROIs and corrected for multiple comparisons across all ROIs.

We also anticipated that, among the theory-of-mind areas, the TPJ might show the clearest evidence of attention type decoding, based on the results of Kelly et al. (2014). Kelly et al. (2014) is particularly relevant to our study because it identified areas showing greater activity when the social attribution of an attentional state to a cartoon face is more difficult, while controlling for factors such as the direction of gaze, emotional valence, and low-level visual features. An important underlying concept in our current work is that modeling someone else’s attention, and processing someone else’s gaze direction, are not the same. Gaze is only one cue to someone else’s attention. In Kelly et al. (2014), we found that the TPJ was not active in association with gaze, and also not active in association with facial expression; but was active in association with a task in which subjects integrated both gaze and expression in order to judge the attentional state of someone else. Considering that the task in Kelly et al. (2014) engaged the TPJ bilaterally, but none of the other theory-of-mind regions investigated here (i.e., STS, precuneus, and MPFC), we anticipated the TPJ might be more likely than the other theory-of-mind ROIs to be involved in encoding attention type. As discussed more in detail in our response to point #4 below, the TPJ region investigated here, which was defined based on studies of theory of mind reasoning, is spatially separated (by 2 cm) from the location of the gaze-following patch within the STS.

To address the reviewers’ concern, we removed the statement in the Introduction that predicts an effect specifically in the TPJ. Instead, in the Introduction, we confine *Prediction 1* to a more cautious statement, predicting that the exogenous-versus-endogenous distinction should be significantly encoded in at least some subset of the ROIs representing the ToM cortical network.

In the revised Discussion section, we added a paragraph (third paragraph) discussing the reasons why the TPJ may have been a special focus of activity here. In that paragraph, we describe more fully why Kelly et al. (2014) is consistent with this finding.

3) Does the main finding reflect a model of others' attention or differences in mental state attribution: sentences describing endogenous attention focus the reader more on the mental act of attending and might also induce further mentalizing (e.g., the reader may wonder: "why would he decide to look for that object?"). In exogenous sentences, the focus of the sentence is instead more strongly on the attention-grabbing object (e.g., "the bright red tie"). Can this alternative interpretation be excluded?

This is an important point that we thought deeply about when constructing the story stimuli. The short answer is that the alternative interpretation can’t be entirely excluded, but we believe it is unlikely. We now address it explicitly in the sixth paragraph of the Discussion section:

“A second, alternative interpretation of the present results is that the exogenous sentences might make the reader focus more on the object in the story (thus engaging less mentalizing), whereas the endogenous sentences might make the reader focus more on the character’s mental act of attending (thus engaging more mentalizing). […] The results point to the left TPJ processing different information in the exogenous and endogenous story types, but not a difference in overall amount of activity.”

4) Searchlight analysis for exogenous v. endogenous attention: Is the cluster centred at -59, -47, 5, labelled left posterior STS (TPJ), really TPJ? It is, after all, squarely in the temporal lobe and some distance (22mm) from the centre of the left TPJ ROI, the latter being in left angular gyrus (areas PGa and PFm). Do you have some independent justification for the labelling of the location of this cluster as TPJ? For example, perhaps it lies within the anterior TPJ (TPJa) subregion identified by Mars et al. (Cerebral Cortex 2012)? The posterior STS area from the searchlight analysis seems to fall into the "gaze-following patch" in the posterior STS as reported by Marquardt et al. 2017. Since this cortical area is not generally considered to be a part of the theory of mind network, the seeming involvement of this area potentially changes the interpretation of the results.

We very much appreciate the reviewers bringing this issue of neuroanatomy to our attention, and for directing us to the Mars et al. (2012) paper. First, we would like to emphasize that the anatomical localization of the Searchlight decoding clusters was based on the projection of the clusters onto sections of the average structural scan generated from the 32 subjects. The projection of volumetric results onto a 3D canonical brain surface is an approximate and imperfect process, and was here used for visualization purposes only. To better clarify the three-dimensional location of the left posterior STS cluster in the endogenous-vs-exogenous Searchlight analysis, we have now included all three sections in the bottom part of Figure 3—figure supplement 3

As can be seen in Figure 3—figure supplement 3, the cluster lies entirely in the posterior section of the STS (and not on the STG, which one might guess from looking at the brain surface projection). As the reviewers point out, the peak of the Searchlight cluster (MNI: -59, -47, 5) is anterior-inferior with respect to the center of our left TPJ ROI (MNI: -52, -56, 24). However, do they belong to the same or different TPJ subregions? According to the connectivity-based TPJ subdivision proposed by Mars et al. (2012), the border between the anterior (TPJa) and posterior TPJ (TPJp) within the posterior STS is somewhere between the MNI y-coordinates -40 and -48 (see Figure 3a in Mars et al.; unfortunately, a more precise definition in terms of MNI coordinates are not reported). Our TPJ ROI is clearly located in TPJp. The Searchlight cluster is likely located in TPJp but very close to the border of TPJa, given that the Searchlight cluster peak is 3 mm closer to the center of gravity for TPJp compared to TPJa (23 mm vs 26 mm). This anatomical labelling is in accordance with another functional TPJ parcellation (Bzdok et al. 2013), in which the TPJp within STS starts at the MNI y-coordinate -47 (see Figure 2 in Bzdok et al. 2013).

It should be noted that both the Mars et al. (2012) and Bzdok et al. (2013) TPJ characterizations are based on the right TPJ. However, the TPJ has a known hemispheric asymmetry regarding functional specialization (Seghier, 2013), neurological lesion effects (Corbetta et al., 2000), functional (Uddin et al., 2010) and anatomical (Caspers et al., 2011) connectivity, as well as cytoarchitectonic borders and gyral pattern (Caspers et al., 2006, 2008). The proposed TPJa and TPJp subdivisions may therefore not directly apply to the left TPJ, and should be interpreted with caution.

Finally, the endogenous-vs-exogenous decoding cluster does not overlap with the gaze-following patch (GFP) reported in Marquardt et al. 2017 (MNI: -55, -67, 6). The GFP is located 20 mm directly posterior to the decoding cluster (and inferior to our TPJ ROI).

We have updated the Supplementary file 7, Figure 3—figure supplement 3, and Supplementary file 1, which now briefly discuss the spatial discrepancy of the left TPJ ROI and the Searchlight decoding cluster. We have also added references to Mars et al. (2012) and Bzdok et al. (2013).

Additional analyses:5) Please report and compare accuracy and RTs for the response to the probe statement. If either differs between conditions, then that becomes a potential confound for the between condition contrasts in the MVPA analyses, considering that the BOLD response to the story and probe events could not be separated.

We thank the reviewers for suggesting this relevant analysis. Among the four social conditions, *self-endo*, *self-exo*, *other-endo*, and *other-exo*, we did not expect any systematic differences in performance on the probe statement task, given the extremely subtle semantic differences across conditions. As predicted, neither the mean accuracies (94.2% vs 93.4% vs 93.1% vs 93.4%; F_3,93_=0.15, p=0.930, repeated-measures ANOVA) nor the mean RTs (1.61s vs 1.60s vs 1.67s vs 1.65s; F_3,93_=1.66, p=0.181, repeated-measures ANOVA) for the response to the probe statement differed significantly across the four conditions.

These behavioral results relating to the response to the probe statement are now included in the revised manuscript, in a new first section of the Results (*Behavioral results*). Note also that since the story period was 10 sec, and the subsequent probe event was 4 sec, it was possible to focus the analysis on the MRI activity evoked by the story, fairly well uncontaminated by the probe event.

6) The TPJ is introduced in the context of theory of mind and social cognition, but it has also been implicated in attentional orienting. Could it be the case that participants simulate such orienting when reading the stories, leading to the above-chance decoding in TPJ? If this is the case, one may similarly expect above-chance decoding in areas that have been implicated in endogenous attention. This should be tested by including dorsal attention network ROIs. Above-chance decoding in such attention regions may inform the interpretation of the TPJ results.

We thank the reviewers for this intriguing suggestion. We have performed the requested analysis and included it in the new submission. If we understand the reviewers’ suggestion correctly, they hypothesize that our subjects, when reading stories, may have simulated exogenous and endogenous attention reorienting by activating the corresponding ventral and dorsal attention networks in a “mirror-neuron-like” fashion. This proposal would explain our TPJ result, because this area is a key node in the ventral attention network and might thus be involved in simulating exogenous attention orienting. If this mechanism is the underlying cause of the TPJ result, one would also predict the involvement of the dorsal attention network when subjects read (and simulated) the endogenous stories.

To test this prediction, as requested, we repeated the endogenous-versus-exogenous decoding analysis in four areas typically considered to constitute the dorsal attention network: the frontal eye fields (FEF), the anterior and posterior intraparietal sulcus (aIPS and pIPS), and the middle temporal complex (MT+). The ROIs were defined as 10-mm-radius spheres around the peak coordinates reported in Fox et al. (2006). The results are shown in Figure 3—figure supplement 6. None of the dorsal attention network ROIs decoded the attention type better than chance (all ps > 0.05, based on permutation testing with 10,000 iterations, uncorrected for multiple comparisons). These findings are thus incompatible with the proposed attention simulation account of the TPJ result.

This issue and these new results are now discussed in the fifth paragraph of the revised Discussion section, and the analysis results are also included in the supplementary material and shown in Figure 3—figure supplement 6.

7) Please provide univariate activity estimates, both in the ROIs and in whole-brain contrasts. This may help to interpret the multivariate results.

The results for the seven univariate contrasts corresponding to the main multivariate analyses are now included in the supplementary material (Supplementary file 5). We report univariate activity corrected for multiple comparisons both at the whole-brain level and within the search volume of each ROI. Notably, there were no significant clusters in any of the ROIs, or anywhere else in the brain, for the *ENDOGENOUS > EXOGENOUS* and *EXOGENOUS > ENDGENOUS* contrasts, not even at the uncorrected threshold p<0.001. These findings are compatible with previous studies (e.g., Hassabis et al. 2009 Current Biology) that have demonstrated the superiority of pattern-sensitive multivariate analyses compared with conventional univariate approaches for detecting differences in activity between conditions with highly similar macroscopic characteristics. In the revised manuscript, we added a sentence in the tenth Discussion paragraph referring to the univariate results.

The SELF > OTHER contrast also did not reveal any significant activity (even at the p<0.001 uncorrected level). The OTHER > SELF contrast revealed one single significant cluster, which was located in the left calcarine sulcus and survived whole-brain correction for multiple comparisons. We speculate that this activation reflects a main effect of the visual input of a name (e.g. “Emma”) in the OTHER condition versus the word “You” in the SELF. We observed no other activations at the p<0.001 uncorrected level in the rest of the brain, and the left calcarine activation was located at a distance from the locations of our ROIs (the STS, TPJ, MPFC, and precuneus) and did not overlap with the decoding results.

The attention type X agent type interactions did not reveal any activity at the whole-brain or ROI levels that survived multiple comparisons.

The Social > Non-Social contrast revealed more significant and widespread activity than the above contrasts, which was expected given the much greater semantic difference between the conditions. We found no significant activity within the ROIs, but four clusters survived correction at the whole-brain level: the left middle orbital gyrus, right SMG, left SFG, and the left insula.

Further support for main finding:8) The reported decoding accuracy values are really quite small and close to 50% (chance), especially for the endogenous vs. exogenous and self vs. other contrasts, even those values that are statistically significant. Please report appropriate effect sizes and the confidence intervals around those effect sizes (for all your reported t-tests, not just those that are statistically significant). It would also be informative if you were to include in your graphs the individual subject data (e.g., mean decoding accuracy per subject for each ROI) and/or plots of the effect sizes and their distributions. For more on effect sizes, their CIs and associated plots, I point you to the following sources and the references therein:https://thenewstatistics.com/itnshttps://thenewstatistics.com/itns/2019/05/20/reply-to-lakens-the-correctly-used-p-value-needs-an-effect-size-and-ci/https://www.estimationstats.com/

We thank the reviewers for this suggestion. We believe the decoding accuracy effects are not so small compared to other findings in the literature. This is especially so when considering that the effects noted by the reviewers are for the most subtle stimulus distinctions in our experiments. The effects are, of course, larger for the social-vs-nonsocial comparisons, because the distinction between the stimuli is much larger. In addition (as suggested by the reviewers in point 10 below), we have now replicated the crucial left TPJ decoding effect using three other anatomical definitions of that theory-of-mind ROI (see new Figure 3—figure supplement 2). All three anatomical definitions of the ROI show a highly significant effect (actually larger than the effect reported in our main analysis in the ROI based on van Veluw and Chance, 2014). Those replications of the finding may help convince the reviewers of the robustness of the result.

We also thank the reviewers for the suggestion to show more information about effect size. In Tables 1 and 2, we had already reported, for all analyses regardless of their significance, the effect size in terms of mean decoding accuracy (relative to chance), the 95% confidence interval around the mean (based on a bootstrap distribution), and the p value (based on permutation testing). To visualize the data more clearly, we have now included two additional supplementary figures featuring violin plots for all other ROI analyses (see Figure 3—figure supplement 1 and Figure 4—figure supplement 2). The violin plots show mean and median decoding accuracy, 95% confidence interval, a kernel density estimation, and individual data points for each decoding analysis and each ROI. (To avoid confusion and visual clutter, we left the figures in the main body of the paper as they were, and presented the new violin plots in the supplementary material.)

9) The authors computed the attentional state decoding separately for "self" and "other". These decoding accuracies did not differ. Please also provide and test the accuracies of self and other separately; this would shed light on the reliability of the main result (which was collapsed across the two conditions) and might also indicate whether the decoding was more reliable (less variable) in self or other.

Thank you for this suggestion. These results are now presented in Figure 3—figure supplement 1 in the revised submission. These results revealed no significant activity in the left TPJ (or in any of the other ROIs), which is likely due to a lack of statistical power (since only half of the data set is used for these analyses). There is, however, an alternative analysis that gets at the same question but does not suffer from a reduction in statistical power, and which does show significant activity in the left TPJ. We included that new analysis in the revision. Please see our response to the related point #11 for a description of that new analysis.

10) Replicate results in one or multiple additional ToM TPJ ROIs. Reviewers raised two suggestions: 1) Surface-based ROI: the TPJ is a highly anatomically variable cortical region that isn't well approximated by a single volume ROI (see Croxson et al. 2017). There are many surface-based ROIs now available that mitigate the effect of this variability, including ones from the authors' own lab. 2) Use a different meta-analysis: The used meta-analysis defines “theory of mind” solely in terms of false-belief tasks (van Veluw and Chance, 2014), which may not be appropriate for delineating the ROIs and their exact locations in your study. Different types of ToM task reliably activate different brain areas, as well as common ones (mPFC and bilateral TPJ: Molenberghs et al., Neuroscience and Biobehavioral Review 2016). Consider using the results of a different meta-analysis, e.g., one that identifies regions based on the conjunction of multiple types of theory-of-mind tasks (e.g., Mar, Annual Review of Psychology 2011; Molenberghs et al., 2016; Schurz et al., Neuroscience and Biobehavioral Reviews 2014).

We welcome the reviewers’ suggestion of replicating the endogenous-versus-exogenous decoding results in different ToM TPJ ROIs. In line with their recommendation, we defined three new left TPJ ROIs based on the peak coordinates reported in Mar (2011), Schurz et al. (2014), and Molenberghs et al. (2016). The results showed significant decoding in the Mar ROI (mean decoding accuracy 53.4%, 95% CI 51.0 to 55.1, p=0.0023), the Schurz ROI (mean decoding accuracy 52.7%, 95% CI 50.8 to 54.4, p=0.0092), and in the Molenberghs ROI (mean decoding accuracy 53.4%, 95% CI 51.3 to 55.2, p=0.0020), suggesting that the attention type decoding in the left TPJ is robust. In the revised paper, these results are included in the supplementary material (see Figure 3—figure supplement 2, also pasted above in relation to major point 8).

The suggestion of re-analyzing the data set using surface-based techniques is interesting and definitely something we aim to pursue in future studies. However, we feel that mixing surface- and volume-based analysis methods, and defining ROIs using fundamentally different approaches, might risk over-complicating the paper. Furthermore, the surface-based TPJ subdivision described by Igelstrom et al. (2016) suffers the same draw-back as the van Veluw and Chance (2014) meta-analysis in that it only involves one type of theory-of-mind task (a false belief task). We therefore decided to use volume-based methods throughout.

11) In previous work (Kelly et al., 2014), the authors were interested in an overlap between attention in self and other. Here, this could be addressed in a cross-decoding analysis, training a classifier on exo vs endo in the "self" stories and testing this on the "other" stories. Above-chance classification in this analysis would strengthen the evidence for lTPJ involvement and would provide additional information that would help interpreting the TPJ findings.

We thank the reviewer for this excellent suggestion. Indeed, one of the goals of the study was to test for differences in the encoding of attention type in self versus others (*Prediction 3*). Attention type could be significantly decoded in the left TPJ when taking all data (“self” and “other” stories) into account (Figure 3A). However, we did not find significant decoding in the left TPJ when analyzing the “self” stories and “other” stories separately (see our response to point #9), which is likely due to a lack of statistical power because the data set was split in two. The absence of a self-vs-other difference in attention decoding could thus be due to either a lack of statistical power, or a true overlap between attention type encoding in self and other.

A direct way of testing for overlap in attention encoding in self and others in the left TPJ, while maintaining statistical power by using the entire data set, is to employ a two-way cross-classification analysis. In line with the reviewers’ suggestion, we have now included this analysis in the paper (Figure 3—figure supplement 7). In each subject, one classifier was trained to discriminate endogenous versus exogenous self-stories and tested on other-stories (using a leave-one-run out approach), and another classifier was trained to discriminate endogenous versus exogenous other-stories and tested on self-stories, from which an average cross-classification decoding accuracy for the left TPJ was obtained. The group-level result shows significant above-chance decoding (mean decoding accuracy 51.9%, 95% CI 49.9 to 54.1, p=0.0393, permutation testing with 10,000 iterations). This finding suggests that there is an overlap in brain mechanisms that participate in the encoding of attention in others and in encoding of attention in the self in the left TPJ.

The new results are now included in the Results section (under *Prediction 3*) and Figure 3—figure supplement 7 in the revised paper. We have also expanded the eighth Discussion paragraph, relating to the interaction between attention type and agent type, to address this point.

[Editors' note: further revisions were suggested prior to acceptance, as described below.]

Essential Revisions:1) Please report and analyze the univariate activity in the ROIs2) More fully report (in main Results section) and discuss the searchlight results.

We thank the editor and two reviewers for their detailed and constructive comments. In the revised submission, we report the univariate activity averaged across all voxels within each ROI, and report and discuss the Searchlight results in the main text. In our view the paper has been greatly improved as a result of these revisions. Below we answer all comments in a point-by-point manner.

Reviewer #1:The authors have addressed many of my concerns. In particular, the additional TPJ ROIs and the demonstration of self-other cross-decoding increased my confidence in the main finding of attention state decoding in lTPJ.One comment has not been sufficiently addressed, however: it would be highly relevant and informative to see the average univariate activity for each of the conditions in the 6 original ROIs, plotted in a graph. The endo>exo contrast should be tested, correcting for the 6 ROIs, similar to how the decoding accuracies are tested (i.e., with the same p<0.05 cut-off). This ROI-based univariate analysis is also interesting for other comparisons, e.g. to test whether the social conditions gave higher activity than the non-social condition, as would be expected based on previous work. Thus, please provide a full univariate analysis on average ROI activity.

We thank the reviewer for this suggestion. In the revised submission, we now include a supplementary figure showing the difference in average univariate activity across all voxels within each ROI, for the endo-vs-exo, self-vs-other, and social-vs-nonsocial contrasts. In line with the voxelwise univariate results reported in Supplementary file 5, none of these contrasts were statistically significant (all FDR-corrected ps > 0.21).

There is a lot of interesting information in the Supplementary figures (e.g., searchlight, additional ROIs), which will not be visible when viewing or printing pdfs. You could consider moving some of this to the main text, or add panels to existing figures where you include some of this information.

We agree with the reviewer, and have incorporated some of the supplementary results into the main text in the revised submission. Specifically, we have added three panels to Figure 3, which now features the additional left TPJ ROI results, the endo-versus-exo Searchlight results, and the cross-classification results.

Reviewer #2:In this revised manuscript, the authors have greatly clarified some key points, most important of which was how well matched the behavior was across all of the conditions. While there is still a potential confound in the social vs. non-social contrast given the difference in reaction time, (a) the confound effects are likely negligible in size, (b) the end result matches the previous literature, and (c) the contrast was only a control analysis and doesn't greatly affect the main point of the paper. However, the handling of the left posterior STS result in the searchlight analysis remains problematic, both in the manuscript and in the author's response.pSTS response and discussion: The main point of the manuscript is to examine whether known theory of mind areas contain information that differentiates between the endogenous vs. exogenous story conditions. The revised manuscript makes quite clear that the answer is yes, activity in the L TPJ differentiates between the two conditions, albeit with a relatively low decoding accuracy. The fact that multiple versions of the L TPJ ROI produces the same result is particularly reassuring. The spotlight analysis, however, finds that there is potentially more discriminatory information to be found in the posterior STS/middle temporal gyrus, and this is where the manuscript runs into problems.In the revised manuscript, this result is simply not mentioned, though it clearly has implications for the interpretations of the result as mentioned in the previous round of reviews. In addition, in the response to the reviewers' comments, the authors give a very convoluted and confusing argument based on MNI coordinates that somehow this focus is somehow (a) part of the TPJ and (b) may be a part of the attention-related TPJa as opposed to their TPJp ROI. In the process of saying that the surface projections are misleading, they state that the focus as falling on the STG on the surface projection, when it is really on the MTG. In addition, they base their arguments on MNI coordinates, but MNI coordinates are notoriously inaccurate in comparing results from different studies, so the author's attempts to justify their conclusions using these coordinates falls flat. The authors own volume images very clearly show the focus as being in the STS and not on either the angular gyrus or supramarginal gyrus that Mars, Corbetta, and others have generally shown the TPJp and TPJa to fall on. And even if this focus was in the TPJa, as the authors seem to hint at, they do not discuss the implications of this result.The authors' defensiveness around this point is frankly puzzling. The searchlight results do not seem to invalidate their main point, only potentially augment it. The fact that the posterior STS (and what seem to be the right FEF and iPCS areas) exhibit higher discriminability between the endo and exo conditions may not be shocking given that the theory of mind areas are likely involved in many operations in this task, whereas the pSTS and right hemisphere areas likely may only be involved in just the imagined attention/sensory processing aspects of the task. It is possible that these areas are "reading out" the differing attentional conditions from the L TPJ. Whatever the explanation may be, the authors seem to be trying to bury this result to shoehorn the results into fitting their a priori model, which is a disservice and misleading to the readers, and needs to be rectified before the manuscript can be published. My recommendation is to at least mention the searchlight results in the main Results section, then add a paragraph to the Discussion discussing the implications of these results. Better yet would be to quantify the discrimination accuracy within each of the foci uncovered in the spotlight analyses.

We are sorry to hear that the reviewer interpreted our previous response as defensive, it was not our intention. And we apologize for erroneously referring to STG in our response (“they state that the focus as falling on the STG on the surface projection”), it should have said MTG, which is obvious from the surface projection in Figure 3—figure supplement 3. In the revised submission, we now include the searchlight results in the main Results section and in the main results Figure 3 (panel C).

First, we would like to take the opportunity put the searchlight result into a broader context. The searchlight analysis is fundamentally different from the ROI analysis. It is not targeted to specific brain areas on the basis of strong predictions. Instead, it is a whole-brain analysis that is much more statistically conservative because of the brain-wide correction for multiple comparisons. In general, one would not necessarily expect the searchlight analysis to align perfectly with the ROI analysis. Its usefulness is that it may reveal any cluster of very strong decoding that was missed by the more sensitive analysis restricted to the ROIs. However, the results of our searchlight analysis revealed no brain-wide significant clusters of decoding for the endogenous versus exogenous distinction. Thus, we cannot make any inferences about attention-type decoding based on the searchlight results. However, in a purely descriptive manner, we report the (four) decoding clusters that survived the conventional uncorrected statistical threshold p<0.001. It is interesting to note that the brain-wide searchlight decoding peak happened to fall in the posterior STS just 20 mm from the center of our predefined left TPJ ROI, and that they belong to the same anatomical subregion of the TPJ (TPJp). In our view, this approximate, descriptive overlap lends at least some additional confidence to our significant left TPJ ROI decoding result. Nevertheless, since the searchlight result consists of an uncorrected decoding map, we have been very careful to not over-interpret the data. We suspect that the reviewer may have taken this caution on our part as defensiveness. Because the searchlight results for the endo-versus-exo distinction were inconclusive, interpretations such as the one offered by the reviewer (“The fact that the posterior STS [and what seem to be the right FEF and iPCS areas] exhibit higher discriminability between the endo and exo conditions may not be shocking given that the theory of mind areas are likely involved in many operations in this task, whereas the pSTS and right hemisphere areas likely may only be involved in just the imagined attention/sensory processing aspects of the task. It is possible that these areas are "reading out" the differing attentional conditions from the L TPJ.”) are in our view therefore highly speculative.

The goal with our previous response was to make an honest attempt at pinpointing the functional location the left pSTS searchlight cluster, which was prompted by the reviewer’s previous comment that questioned whether the cluster belong to the TPJ at all and speculated that it perhaps belongs to TPJa or coincides with the gaze-following patch reported in Marquardt et al. 2017. In contrast to the reviewer’s assertion that the TPJa and TPJp encompass only the angular gyrus and the SMG, the connectivity-based subdivision of TPJa and TPJp described by Bzdok et al. (2013) and Mars et al. (2012) both define the pSTS as part of the TPJ. According to this definition, our pSTS searchlight cluster clearly belongs to the TPJ and specifically to the posterior subregion TPJp. The reviewer appears to have the impression that we are arguing the cluster belongs to the TPJa (“the authors [argue that] this focus is somehow (a) part of the TPJ and (b) may be a part of the attention-related TPJa as opposed to their TPJp ROI.” and “And even if this focus was in the TPJa, as the authors seem to hint at […]”); however, we are not. Furthermore, the pSTS searchlight cluster does not coincide with the gaze-following patch reported in Marquardt et al. 2017.

To help clarify the relationship between the left TPJ ROI, in which we obtained significant decoding, and the left STS, in which we found a non-significant peak in the searchlight analysis, we performed a further analysis for the reviewer. In this new analysis, we used a more lax statistical threshold, to better examine whether there is any halo of activity hidden under the statistical threshold. Instead of using the arbitrary uncorrected threshold of p<0.001, we used the uncorrected threshold of p<0.01, in the searchlight analysis. Author response image 1 shows the result. Using this threshold, the left posterior STS cluster merged with clusters encompassing both the angular gyrus, the SMG and the MTG. The general area of activity is now more clearly in the TPJ, including the TPJ ROI and extending into the STS. It is important to keep in mind, however, that all of these methods – the method that shows a small peak in the STS, and the method that shows a large cluster encompassing the TPJ – should not be taken as definite, statistically strong findings. They are the result of exploratory searchlight analysis and, though interesting, should be taken cautiously.

We certainly want to avoid the impression of “burying” the searchlight results. Although descriptive, they are interesting and compatible with our ROI results, and constitute valuable information not least for future studies. Therefore, we now include the endo-versus-exo searchlight results in the updated main results figure (Figure 3C) and report the results, including the decoding accuracy for the pSTS peak voxel (53.7%), in the main Results section under “Prediction 1”.